# CAN VISION-LANGUAGE MODELS LEARN MEDICINE FROM PUBLIC EDUCATIONAL VIDEOS?

## ABSTRACT

Publicly available biomedical videos, such as those on YouTube, serve as valuable educational resources for medical students. Unlike standard machine learning datasets, these videos are designed for human learners, often mixing medical imagery with narration, explanatory diagrams, and contextual framing. In this work, we investigate whether such pedagogically rich, yet non-standardized and heterogeneous videos can effectively teach general-domain vision-language models biomedical knowledge. To this end, we introduce OpenBiomedVid, a biomedical video instruction tuning dataset comprising 1031 hours of video-caption and Q/A pairs, curated through a multi-step human-in-the-loop pipeline. Diverse biomedical video datasets are rare, and OpenBiomedVid fills an important gap by providing instruction-style supervision grounded in real-world educational content. Surprisingly, despite the informal and heterogeneous nature of these videos, the fine-tuned Qwen-2-VL models exhibit substantial performance improvements across most benchmarks. The 2B model achieves gains of 98.7% on video tasks and 71.2% on image tasks. The 7B model shows improvements of 40.5% on video and 11.2% on image tasks compared to their respective base models. To address the lack of standardized biomedical video evaluation datasets, we also introduce two new expert curated benchmarks, MIMICEchoQA and SurgeryVideoQA. On these benchmarks, the 2B model achieves gains of 99.1% and 98.1%, while the 7B model shows gains of 29.3% and 52.1%, respectively, demonstrating the models' ability to generalize and perform biomedical video understanding on cleaner and more standardized datasets than those seen during training. These results suggest that videos created for human learning offer an effective training signal for biomedical VLMs.

## 1 INTRODUCTION

Vision-language models (VLMs) have made significant progress in integrating visual and textual modalities, achieving strong performance in image captioning, visual question answering, and multimodal reasoning (Liu et al., 2023; Meta, 2024; Hurst et al., 2024; Beyer et al., 2024; Wang et al., 2024; Team et al., 2024; Wang et al., 2025a). Recent advancements in video-language models have further extended these capabilities to dynamic visual data, enabling the understanding of temporal dependencies in videos (Beyer et al., 2024; Wang et al., 2024; Team et al., 2024).

In the biomedical domain, there is a growing interest in applying VLMs to tasks such as medical report generation, disease classification, and question answering (Saab et al., 2024; Thapa et al., 2024; Tu et al., 2024; Li et al., 2023a). However, the potential of these models for biomedical video understanding remains largely unexplored, primarily due to the limited availability of diverse biomedical video instruction tuning datasets in the public domain, which are essential for fine-tuning.

While publicly available biomedical image datasets have supported VLM research, biomedical video datasets remain limited in both scale and diversity. Existing resources are typically narrow in scope. For example, MedVidQA (Gupta et al., 2023), though also derived from YouTube, emphasizes consumer-facing health topics and supports localization and segmentation tasks rather than open-ended question answering. AVOS (Goodman et al., 2024) focuses on surgical videos, with annotations for tool tracking, procedural step recognition, and skill assessment via motion kinematics, but does not address broader biomedical education or instruction-tuning. NurViD (Hu et al., 2023) centers exclusively on nursing procedures, MIMIC-IV-ECHO (Gow et al., 2023) is restricted to

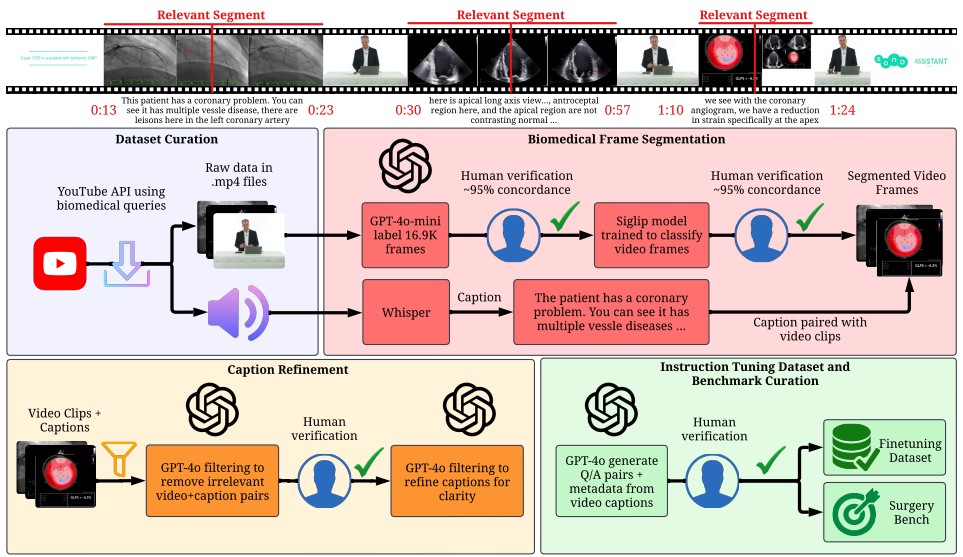

Figure 1: Data generation pipeline. (1) **Curation**: YouTube videos are collected with clinically guided queries. (2) **Frame Segmentation**: A SigLIP model fine-tuned on GPT-labeled data filters biomedical frames. (3) **Caption Refinement**: Whisper transcriptions are cleaned with GPT-4o for accuracy and grounding. (4) **Instruction Generation**: GPT-4o produces multi-turn Q/A pairs and metadata. Human experts verify quality throughout.

echocardiography, and the EndoVis Challenge datasets (Nwoye et al., 2023) primarily support surgical tool segmentation (see Section B for extended discussion).

These limitations highlight the need for a comprehensive biomedical video instruction tuning dataset that covers a diverse range of biomedical concepts, including human anatomy, disease pathology, medical procedures, and clinical diagnostics. The lack of standardized biomedical video evaluation datasets further hinders progress, making it challenging to effectively benchmark model performance.

In recent years, YouTube has become an important resource for medical education (Osman et al., 2022; Akakpo & Akakpo, 2024; Chen et al., 2019; Derakhshan et al., 2019; Rapp et al., 2016), hosting a wide range of biomedical videos on anatomy, surgical procedures, diagnostic techniques, and clinical case discussions. Unlike institutional datasets, these videos are informal and heterogeneous, blending real-world medical imagery with narration, diagrams, and didactic commentary. Despite this, they have successfully supported the training of thousands of students and practitioners. This motivates our central question: *Can open-domain VLMs learn meaningful biomedical vision concepts from publicly available educational videos?* Answering this is key to understanding whether pedagogically oriented content can also provide effective supervision for biomedical AI.

To this end, we introduce OpenBiomedVid, a 1,031-hour collection of biomedical educational videos from YouTube, curated through a multi-step human-in-the-loop process. The dataset spans multiple domains—including cardiology, radiology, and surgery—and covers diverse anatomical regions such as the cardiac, vascular, musculoskeletal, and head and neck systems. To address the lack of biomedical video evaluation resources, we further release two expert-curated benchmarks: SurgeryVideoQA, with 2,692 QA pairs from high-quality surgical videos, and MIMICEchoQA, with 622 QA pairs derived from MIMIC-IV-ECHO (Gow et al., 2023) focusing on echocardiography.

Using this dataset, we fine-tune Qwen-2-VL models and demonstrate substantial performance gains across image, and video benchmarks compared to baseline models. Our results suggest that open-domain VLMs can indeed learn meaningful biomedical concepts from publicly available videos.

**Our contributions are:**

- We curate OpenBiomedVid, a 1,031-hour biomedical video-text dataset from publicly available YouTube content. Unlike traditional datasets, it consists of educational material

combining medical imagery with narration and explanations, spanning diverse domains and anatomical regions, and built through a human-in-the-loop pipeline.

- We introduce two expert-curated benchmarks, SurgeryVideoQA and MIMICEchoQA, to evaluate VLMs on surgical and echocardiographic understanding via Q/A, addressing the lack of standardized biomedical video evaluation.

- We fine-tune `Qwen2-VL` and `InternVL3` models at multiple scales on OpenBiomed-Vid, showing substantial gains on video and image benchmarks and demonstrating the effectiveness of educational videos as supervision signals for biomedical VLMs.

## 2    Biomedical Video Dataset

In this section, we describe our end-to-end pipeline of collecting raw video data to generate fine-tuning and evaluation datasets, as shown in Figure 1.

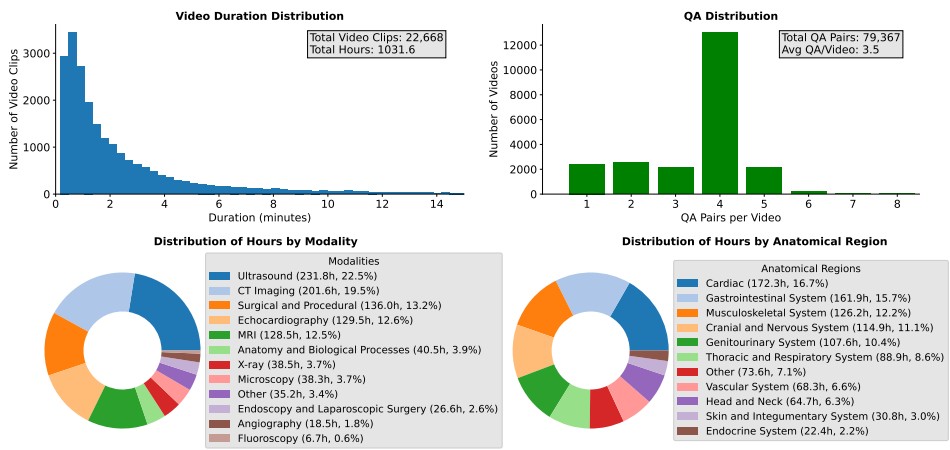

Figure 2: Distribution of the fine-tuning dataset across different biomedical modalities and anatomical regions.

### 2.1    Dataset Curation

We curated our dataset by collecting publicly available biomedical videos from YouTube. The process involved collaboration with a clinician to compile a list of relevant search queries across multiple biomedical video modalities, such as echocardiograms, surgical procedures, ultrasounds, angiograms, colonoscopies, endoscopies, and laparoscopies. Additionally, we manually searched for biomedical-related YouTube channels and included all videos from those channels in our search list. Using this combination of expert-guided and manually inspected queries, we programmatically identified and retrieved videos. After removing duplicates, our final raw dataset consisted of 24,560 videos, totaling 4,137 hours of content.

### 2.2    Data Processing

**Biomedical Frame Segmentation.**    We randomly sampled 16.9K frames from the entire raw dataset and annotated them using `GPT-4o mini` (Hurst et al., 2024), classifying each frame as either biomedical or non-biomedical (prompt in Section F.1). The annotation prompt was iteratively refined to ensure accurate labeling, prioritizing frames that contained visual biomedical features such as medical scans, surgical procedures, and anatomical visualizations, while excluding irrelevant content, including text overlays or images of medical professionals without supporting visual data. An expert human annotator independently labeled 1,000 frames, achieving a 95.0% agreement with `GPT-4o mini`. We leveraged this labeled data to fine-tune `google/siglip-base-patch16-224` classifier (Zhai et al., 2023). The resulting model, `siglip-medical`, achieved 95.48% agreement with human annotations on a benchmark of 500 manually labeled medical images.

We applied a segmentation algorithm based on a sliding window approach to identify high-confidence biomedical segments. Using the fine-tuned `siglip-medical` model, we classified frames at 0.5-second intervals for all raw videos, generating a probability distribution across the entire video length. Frames with probabilities above 0.64 were classified as biomedical, while those below this threshold were labeled non-biomedical. Consecutive biomedical frames were grouped into coherent segments, allowing gaps of up to 10 non-biomedical frames to ensure smooth segmentation without compromising accuracy.

We then paired these segments with transcriptions generated by `openai/whisper-large-v3` (Radford et al., 2023) to create video-caption pairs. This process resulted in a refined dataset comprising approximately 45,536 video clips and 1,600 hours of biomedical content.

**Caption Refinement.** To enhance caption quality, we implemented a two-step filtering and standardization process using `GPT-4o` (Hurst et al., 2024). The first step involved removing ambiguous or purely textual captions, retaining only those that described observable biomedical content, such as procedures, diagnoses, and anatomical demonstrations. This step reduced the dataset to 22,668 captions and 1031.6 hours of videos.

The second step focused on improving clarity and consistency by standardizing biomedical terminology, preserving spatial and temporal context, and ensuring that descriptions were concise yet informative. Redundant or overly colloquial language was removed, and captions were refined to maintain an objective and neutral tone (Section F.2).

**Instruction Tuning Dataset and Metadata Generation.** We processed the cleaned captions using `GPT-4o` to generate both question-answer (Q/A) pairs and metadata annotations, including video modality and anatomical regions (Sections F.3 and F.4). The Q/A pairs encompassed a range of tasks, such as hierarchical questions progressing from general to specific, temporal and sequential reasoning, comparative and explanatory queries, as well as diagnostic and procedural understanding.

We then manually reviewed and divided the dataset into fine-tuning and evaluation set, ensuring that all video segments from a single video were contained within a single split to prevent data leakage. The evaluation set was focused specifically on surgical videos, which we discuss further in Section 2.3.

Figure 2 shows the distribution of our fine-tuning dataset, which comprises 22,668 video clips totaling 1031.6 hours of biomedical content. Most videos are under 5 minutes long, with an average length of 2 minutes. In total, we generated 79,367 Q/A pairs, averaging 3.5 Q/A pairs per video clip. The most common video modalities include ultrasound (231.8h), CT imaging (201.6h), and surgical and procedural (136.0h).

Similarly, the predominant anatomical regions in our fine-tuning dataset are the cardiac system (172.3h), gastrointestinal system (161.9h), musculoskeletal system (126.2h), and cranial and nervous system (114.9h). Appendix Figure S7 presents the resolution distribution of the fine-tuning dataset.

We provide some qualitative examples of our fine-tuning dataset in Section G. To illustrate the qualitative differences between our fine-tuning and evaluation datasets, we present side-by-side comparisons of representative samples in Figure 3.

## 2.3 Video Evaluation Dataset Curation

There is a lack of biomedical video benchmarks for evaluating large VLMs. Most of the datasets are focused primarily on classification. This motivated us to curate two relevant biomedical video evaluation datasets, which we describe in detail below.

**SurgeryVideoQA.** This benchmark evaluates VLMs in the domain of open surgery. Although curated from the same YouTube source pool as our fine-tuning dataset, SurgeryVideoQA is substantially cleaner due to extensive expert review. A human expert manually inspected all videos to exclude those with overlaid text, embedded answers, or irrelevant segments, retaining only clips in which the majority of content was directly related to surgical procedures. Through this process, we identified 471 high-quality videos totaling 21.9 hours of footage. Importantly, there is no overlap between training and evaluation: we filtered strictly by unique video ID to ensure that no video appearing

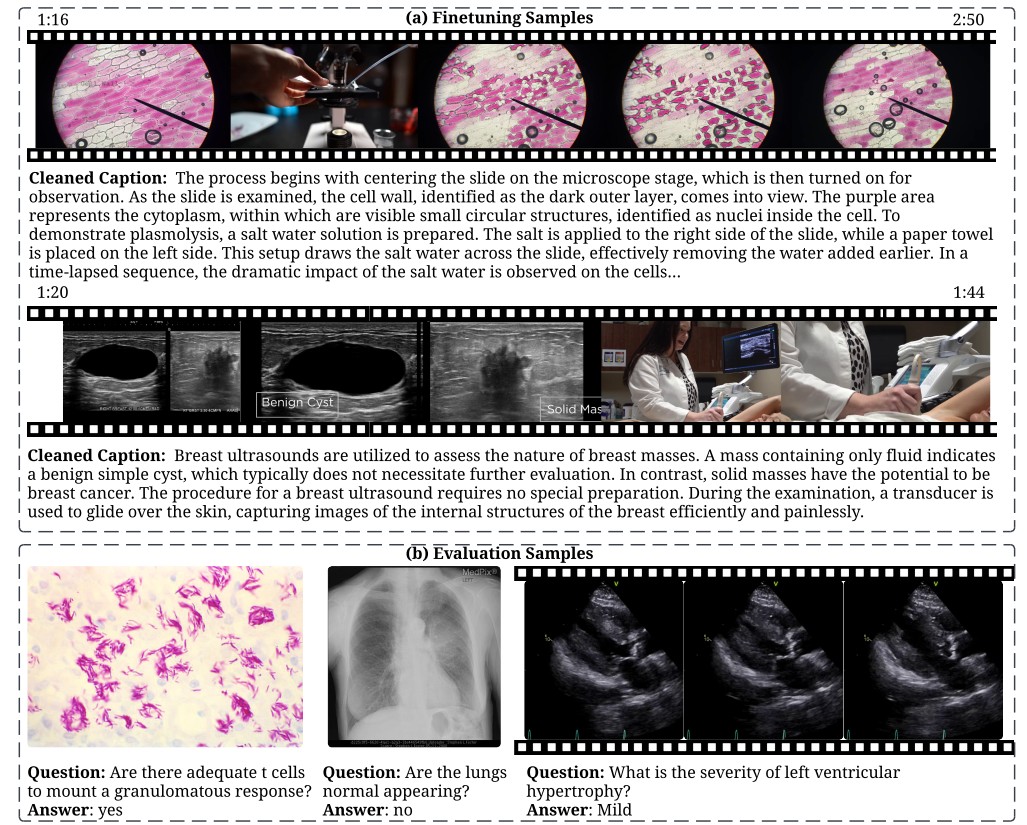

Figure 3: Comparison between the fine-tuning dataset and the evaluation dataset, highlighting the distribution shift. The evaluation dataset is significantly cleaner and more structured than the fine-tuning dataset, enabling a more accurate assessment of model performance.

in the evaluation set was used for training. This separation was further verified through manual inspection.

Using the cleaned captions associated with these videos, we generated multi-turn Q/A pairs with GPT-4o (prompt in Section F.5). We optimized prompt for generating questions that assess a model's understanding of procedural steps, anatomical structures, surgical tools, and intraoperative decision-making. The Q/A format consists of concise, open-ended answers rather than multiple-choice options, providing a more challenging and realistic setting for biomedical applications. A medical doctor then manually reviewed all videos and their corresponding Q/A pairs to remove any trivial or non-relevant questions.

Figure S6 presents the distribution of the SurgeryVideoQA dataset. The dataset contains 2,692 Q/A pairs, with an average of 5.7 pairs per video. The predominant anatomical regions are the gastrointestinal system (5.9 hours, 141 videos), head and neck (5.2 hours, 100 videos), and the skin and integumentary system (3.8 hours, 79 videos). Qualitative examples of the SurgeryVideoQA dataset are provided in Section G.2. The distribution of video resolutions in the SurgeryVideoQA dataset is shown in Appendix Figure S8.

**MIMICEchoQA.** This benchmark is derived from a publicly available echocardiogram video dataset called MIMIC-IV-ECHO (Gow et al., 2023). We paired each study in the dataset with the nearest available discharge summary following the video, provided the time difference between the study date and discharge date was within 7 days, removing studies without discharge summary within the timeframe. From the matched discharge summaries, we extracted the transthoracic echocardiography (TTE) and ECHO sections as proxies for cardiologist reports. These sections typically contain diagnostic information such as ejection fraction and cardiac abnormalities.

To standardize the input format, we converted each DICOM file into an `.mp4` video. The corresponding cleaned reports were then processed using `Qwen-2-72B-Instruct` to generate multi-turn, closed-ended Q/A pairs. However, this automated process occasionally produced questions referencing anatomical structures not visible in the associated videos. To mitigate this, we employed a view classification model (Vukadinovic et al., 2024) to label each video with its specific echocardiographic view (e.g., A3C, A4C), allowing us to filter out unanswerable questions based on view-specific visibility constraints.

Because the view classifier is not perfectly accurate, and to ensure clinical validity, two board-certified cardiologists manually reviewed the generated Q/A pairs. This review identified and removed questions that remained unanswerable given the visual content of the videos, even after automated filtering. This process resulted in a final set of 620 high-quality, clinically valid Q/A pairs. The prompt used for Q/A generation is provided in Section F.6, and qualitative examples from the dataset are shown in Section G.3.

# 3 Experiments and Results

## 3.1 Training

Having curated both the fine-tuning and evaluation datasets, we set out to investigate a core question: *Can open-domain vision-language models learn medicine by studying publicly available educational biomedical videos?* To explore this, we adopted the Qwen2-VL model series as our primary backbone—specifically `Qwen2-VL-2B-Instruct` and `Qwen2-VL-7B-Instruct` (Wang et al., 2024)—given their strong performance among open-source vision-language models and their ability to process both images and videos. For completeness, we also report results from larger models such as `Qwen2-VL-72B-Instruct`, a general-domain multimodal model (`InternVL3-8B`), and closed-source systems including `Gemini-2.0-Flash` and `GPT-4o`, in the ablation study (Section 4).

Our fine-tuning dataset comprises both video-caption pairs and Q/A pairs. This approach ensures that the model is exposed to the same data in different formats, enhancing its ability to learn from diverse input structures. We fine-tuned both the 2B and 7B models on a single node with 8 NVIDIA H100 GPUs for one epoch, tuning the adapter layers and the language model. The fine-tuning process took five hours. Additional training details, including key hyperparameters, are provided in Section E.

**Baselines.** Since this study aims to assess whether fine-tuning on publicly available educational biomedical videos improves a model's medical understanding, our baselines are the `Qwen2-VL-2B-Instruct` and `Qwen2-VL-7B-Instruct` models prior to fine-tuning.

**Evaluation.** Given the open-ended nature of SurgeryVideoQA, direct string matching was unsuitable for accuracy measurement. Instead, we employed `GPT-4o` as an automatic judge, comparing model responses against reference answers and assigning binary scores (1 for mostly correct, 0 for incorrect). The evaluation prompt is provided in Section F.7. Because `GPT-4o` was also used during data cleaning and curation, there is a potential risk of stylistic bias. To assess this, we conducted an additional evaluation with `Gemini-2.0-Flash` as an independent judge, with results discussed in the ablation study (Section 4). For MIMICEchoQA and other text- and image-based tasks, which use closed-ended Q/A, we measured performance through direct string matching and accuracy.

## 3.2 Main Results

We evaluated our fine-tuned models across three categories: video benchmarks, image benchmarks, and text-only benchmarks, to assess their holistic biomedical capabilities.

**Video Benchmarks.** Figure 4 (a) shows that fine-tuning on biomedical video-text data led to substantial improvements over the baseline models. On MIMICEchoQA, the accuracy of the `Qwen-2-VL-2B-Biomed` model increased from 21.1% to 42.0%, representing an 99.1% relative improvement, while the `Qwen-2-VL-7B-Biomed` model improved from 37.9% to 49.0%, a 29.3% gain. On SurgeryVideoQA, the `Qwen-2-VL-2B-Biomed` model improved from 10.3% to 20.4%, a 98.1% relative increase, while the `Qwen-2-VL-7B-Biomed` model rose from 16.5% to 25.1%, marking a 52.1% improvement.

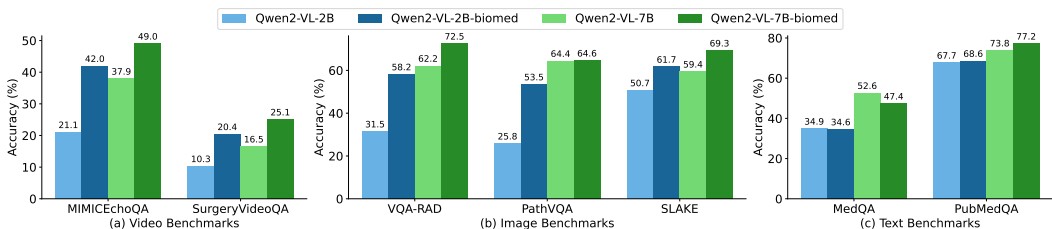

Figure 4: Comparison of fine-tuned models and baseline models across video, image, and text benchmarks.

Since no publicly available biomedical video-language models currently exist, we evaluated `GPT-4o` on both MIMICEchoQA and SurgeryVideoQA. Interestingly, our fine-tuned `Qwen-2-VL-7B-Biomed` model outperformed `GPT-4o` on MIMICEchoQA (49.0% vs. 41.6%), while `GPT-4o` achieved the highest performance on SurgeryVideoQA (35.8%), surpassing all other models. Table 1 summarizes the performance of additional open-source multimodal models.

Despite these improvements, performance on video benchmarks remains significantly lower than on text and image benchmarks (shown below), even after fine-tuning. This suggests that the heterogeneous nature of biomedical videos—characterized by complex procedures, dynamic imaging, and variations in quality—poses significant challenges. Additionally, noisy captions and limited alignment between video content and text further complicate comprehension. The open-ended evaluation for SurgeryVideoQA, where models generate free-form responses scored by `GPT-4o`, also makes these tasks inherently more challenging compared to the closed-ended evaluations used for text and image benchmarks.

Further analysis of dataset scaling, provided in Appendix C, shows that while image understanding saturates quickly, video performance continues to improve as more training data is added. This highlights both the data-hungry nature of video-language learning and the potential benefits of expanding biomedical video resources.

**Image Benchmarks.** For biomedical image understanding, we evaluated the models on three widely used benchmarks: PathVQA (He et al., 2020), VQA-RAD (Lau et al., 2018), and SLAKE (Liu et al., 2021). PathVQA is a visual question-answering dataset focused on pathology images, testing models on histopathological structures and diagnostic reasoning. VQA-RAD is a radiology-based dataset containing X-rays, CT scans, and MRIs, requiring models to interpret medical imaging and answer domain-specific questions. SLAKE is a multimodal VQA dataset that spans various medical imaging modalities, covering questions on anatomy, clinical conditions, and diagnostic interpretation.

We focused on closed-ended question evaluation and measured model performance using accuracy. Figure 4 (b) presents a comparison of fine-tuned models against their respective baselines. On VQA-RAD, the `Qwen-2-VL-2B-Biomed` model achieved a 84.8% improvement over the baseline, while the `Qwen-2-VL-7B-Biomed` model showed a 16.6% improvement. On PathVQA, the `Qwen-2-VL-2B-Biomed` model outperformed the baseline by 107.4%, and the `Qwen-2-VL-7B-Biomed` model improved by 0.3%. On SLAKE, we observed a 21.7% improvement for the `Qwen-2-VL-2B-Biomed` model and a 16.7% gain for the `Qwen-2-VL-7B-Biomed` model. For additional context, we also report results from a biomedical VLM, `Dragonfly-Med` (Thapa et al., 2024) (78.1% on VQA-RAD, 90.6% on PathVQA, and 91.6% on SLAKE).

To further analyze performance across different imaging modalities, we stratified the images from SLAKE and VQA-RAD into three categories: X-ray, CT scans, and MRI. The results are shown in Appendix Figure S5. Across all modalities, fine-tuned models significantly outperformed baselines. The highest accuracy was achieved on MRI images, followed by CT scans, which aligns with the distribution of our training data (Figure 2). Notably, despite having only 38.5 hours of X-ray training data, the model performed competitively, suggesting that fine-tuning contributed substantially to its improvement over the base model.

**Text Benchmarks.** We evaluated our models on two widely used biomedical text benchmarks: MedQA (Jin et al., 2021) and PubMedQA (Jin et al., 2019). MedQA is a multiple-choice question answering dataset designed to assess medical knowledge, including USMLE-style exam questions. PubMedQA consists of biomedical research abstracts paired with clinical questions, requiring models to draw conclusions based on scientific literature.

Figure 4(c) presents the results on these text benchmarks. In contrast to the substantial gains observed in video and image benchmarks, improvements on text benchmarks are less consistent, and in some cases, performance slightly declines. For instance, on MedQA, the performance of `Qwen-2-VL-2B-Biomed` drops marginally from 34.9% to 34.6%, and `Qwen-2-VL-7B-Biomed` sees a more notable decline from 52.6% to 47.4%. On the other hand, we observe modest gains on PubMedQA: `Qwen-2-VL-2B-Biomed` improves from 67.7% to 68.6%, and `Qwen-2-VL-7B-Biomed` improves from 73.8% to 77.2%. For reference, we include results from similarly sized models trained on large-scale biomedical corpora, such as Meerkat-7B (Kim et al., 2024) for MedQA (70.6%) and AntGLM-Med (Li et al., 2023c) for PubMedQA (80.6%).

| Model | MIMICEchoQA | SurgeryVideoQA | Average |
|---|---|---|---|
| Video-ChatGPT (Maaz et al., 2024) | 31.7 | 7.9 | 19.8 |
| Video-LLaVA (Lin et al., 2024) | 32.0 | 9.2 | 20.6 |
| Phi-3.5-vision-instruct (Abdin et al., 2024) | 41.1 | 8.5 | 24.8 |
| Phi-4-multimodal-instruct (Abouelenin et al., 2025) | 37.8 | 8.5 | 23.2 |
| InternVideo2.5-Chat-8B (Wang et al., 2025b) | 40.3 | 16.4 | 28.4 |
| Qwen2-VL-7B-Instruct (Wang et al., 2024) | 37.9 | 16.5 | 27.2 |
| Qwen2.5-VL-7B-Instruct (Bai et al., 2025) | 34.0 | 17.9 | 25.9 |
| Qwen2-VL-72B-Instruct (Wang et al., 2024) | 37.5 | 24.4 | 31.0 |
| Qwen2.5-VL-72B-Instruct (Bai et al., 2025) | 34.2 | 26.2 | 30.2 |
| Gemini-2.0-Flash (Team et al., 2023) | 38.4 | 26.8 | 32.6 |
| GPT-4o (Hurst et al., 2024) | 41.6 | 35.8 | 38.7 |
| o4-mini (OpenAI, 2025) | 43.9 | **46.6** | **45.3** |
| Qwen2-VL-2B-biomed | 42.0 | 20.4 | 31.2 |
| Qwen2-VL-7B-biomed | **49.0** | 25.1 | 37.1 |

Table 1: Performance of open-source and proprietary multimodal models on two biomedical video QA benchmarks: MIMICEchoQA and SurgeryVideoQA. Models generally perform better on MIMICEchoQA, while SurgeryVideoQA remains more challenging. Fine-tuning on our biomedical video dataset (Qwen2-VL-biomed) yields consistent improvements.

# 4 Ablations

**Generalizability of OpenBiomedVid Across Multimodal Architectures.** Our initial experiments focused on Qwen2-VL due to its strong open-source baseline, native video support, and reproducibility. To assess whether OpenBiomedVid benefits other architectures, we additionally fine-tuned InternVL3-8B, another widely used vision-language model with native video capabilities. As shown in Supplementary Table S2, InternVL3-8B-Biomed achieves consistent improvements across both video and image-based benchmarks.

We also examined scaling effects by fine-tuning the larger `Qwen2-VL-72B`. Despite its strong baseline, fine-tuning with OpenBiomedVid further improved performance across tasks, particularly on biomedical video benchmarks. These results demonstrate that OpenBiomedVid provides meaningful gains even at the 72B scale. Overall, OpenBiomedVid is architecture-agnostic: it enhances performance across different model families and scales, confirming its broad utility for biomedical vision-language learning.

**Benchmarking Public Multimodal Models on Biomedical Video Benchmarks.** We evaluate a range of open-source and proprietary multimodal models on MIMICEchoQA and SurgeryVideoQA. Some models—such as Qwen variants and InternVideo—support native video input, while others (e.g., Phi and GPT models) rely on multi-image inputs by sampling and processing up to 250 frames uniformly.

On MIMICEchoQA, which features short echocardiogram clips lasting only a few seconds, most models—including those without native video support—perform reasonably well. The brevity and focused nature of these videos allow frame-based methods to extract sufficient information for accurate question answering.

In contrast, SurgeryVideoQA is a more demanding benchmark: surgical videos are long, complex, and require fine-grained temporal reasoning. Architectural differences become clearer here—models like Phi, which treat frames independently, perform far worse than video-native models such as InternVideo and Qwen. Proprietary systems (`GPT-4o`, `o4-mini`) achieve the highest scores, even surpassing our fine-tuned models. Because `GPT-4o` was also used during caption refinement, one concern is potential stylistic bias. To test this, we repeated the evaluation with `Gemini-2.0-Flash` as an independent judge. Rankings remained consistent across both evaluators (Supplementary **??**), suggesting that the performance gap reflects genuine modeling differences rather than alignment artifacts.

Our instruction-tuned models, `Qwen2-VL-biomed`, achieve strong gains on both benchmarks, outperforming most baselines. This demonstrates the value of domain-specific fine-tuning and narrows the gap to proprietary systems. Nonetheless, while short-form biomedical videos are well-handled, performance on long-form surgical content remains limited, highlighting the need for more advanced architectures and training strategies that can reason over extended temporal contexts.

# 5    Discussion and Conclusion

We introduce OpenBiomedVid, the first large-scale and diverse instruction-tuning dataset for biomedical video-language modeling, together with two standardized benchmarks, SurgeryVideoQA and MIMICEchoQA, spanning procedural and diagnostic modalities. Despite their informal format, these videos yield consistent gains on video and image benchmarks highlighting their value as an underutilized multimodal training resource.

LLM involvement in caption refinement and Q/A generation introduces potential risks of hallucination. To mitigate this, we incorporated human experts at multiple stages of dataset construction and benchmark creation. We also designed Q/A prompts to preserve medical semantics while minimizing stylistic rewriting. While some residual hallucinations may remain, models trained on OpenBiomedVid demonstrate gains on external standardized benchmarks such as MedQA and PubMedQA, indicating alignment with clinically validated knowledge.

To ensure semantic and pedagogical fidelity, we applied several quality controls: curating videos from trusted educational channels, filtering with a biomedical frame classifier, and performing large-scale entity recognition that identified over 260,000 high-confidence biomedical entities in the Q/A corpus. Our evaluation benchmarks were constructed or vetted by medical doctors, providing clinically grounded assessment tasks. These safeguards enhance the dataset's reliability while leaving room for future work to further align with standardized curricula and textbooks.

Despite improvements, our fine-tuned models remain far from clinically reliable. Video-language understanding in medicine is still at an early stage. Future work should integrate video-based training with curated image and text corpora and conduct quantitative hallucination analysis. Importantly, models trained on OpenBiomedVid must not be used for clinical decision-making; their intended purpose is advancing research on multimodal medical reasoning.

Working with publicly available YouTube videos raises important questions about data privacy, content licensing, and downstream safety. We deliberately release only video URLs and derived annotations (captions, QA pairs), without redistributing video content, and comply with the PhysioNet Data Use Agreement for MIMIC-IV-ECHO. Nevertheless, incidental PHI may exist in public videos, and model misuse in clinical settings poses safety risks. We provide a full discussion of these issues and our mitigation strategies in the Ethics Statement (Section A).

In summary, our study lays the groundwork for biomedical video-language modeling by curating a large-scale instruction-tuning dataset and introducing two clinically meaningful benchmarks. We hope this resource catalyzes further research into multimodal medical understanding, while encouraging the community to address open challenges in data quality, ethical use, and clinical grounding.

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

# Appendix Contents

# A  Ethics Statement

Our work raises several ethical considerations related to data privacy, consent, copyright, and downstream safety.

*Data sources and licensing.* OpenBiomedVid was constructed from publicly available biomedical educational videos on YouTube. To respect content creators' rights and YouTube's terms of service, we do not redistribute any video files or thumbnails. Instead, we release only video URLs and derived metadata (captions, QA pairs). All dataset and model releases are restricted to non-commercial, research-only use.

*Privacy and PHI risk.* Although YouTube prohibits the posting of protected health information (PHI), some videos may still contain incidental identifiers such as faces, voices, or on-screen text. We acknowledge this residual risk. To mitigate it, we curated videos primarily from trusted educational channels, applied automated biomedical frame filtering (95% agreement with human annotators), and incorporated human expert review. Downstream users are cautioned that incidental PHI may remain in linked content. If such cases are identified, we will promptly notify YouTube and remove affected entries.

*Consent and attribution.* All content used in this work was publicly available. This practice is consistent with prior academic works that curate datasets from YouTube and release only URLs and annotations (e.g., MiraData (Ju et al., 2024), Mr. HiSum (Sul et al., 2023), AVOS (Goodman et al., 2024)). To promote transparency, we will include a credit list of source YouTube channels in our dataset documentation.

*Use of large language models.* LLMs (primarily GPT-4o) were used for caption refinement, metadata annotation, and Q/A generation, with human-in-the-loop validation by medical experts. GPT-4o was also used as an evaluator on SurgeryVideoQA, though we verified robustness by repeating evaluation with an independent judge (Gemini-2.0-Flash), which yielded consistent rankings. While LLM use raises risks of hallucination and stylistic bias, our safeguards and external evaluations mitigate these concerns.

*Downstream safety.* The models trained in this study are not intended for clinical deployment. They are trained on noisy, heterogeneous educational content and evaluated on open-ended QA tasks. Even the strongest models remain far from perfect accuracy. We explicitly warn that misuse of these models for clinical decision-making could pose safety risks. All dataset and model releases will include a prominent disclaimer: *"This model is intended for research use only. It is not clinically validated and must not be used for medical diagnosis or treatment."*

*Compliance.* For MIMIC-IV-ECHO, we strictly adhered to PhysioNet's Data Use Agreement. For YouTube data, we complied with the platform's terms by releasing only URLs and annotations, not video files. All resources will be distributed under a non-commercial research license.

In summary, we acknowledge the ethical complexity of curating biomedical datasets from public sources. By releasing only metadata, involving medical experts, and explicitly warning against clinical use, we aim to advance research while minimizing risks. We welcome community feedback and contributions to further strengthen privacy protections and safety safeguards.

# B  Related Work

## B.1  Vision-Language Models

Vision-Language Models (VLMs) integrate visual and textual modalities for tasks such as image captioning, visual question answering, and multimodal reasoning (Li et al., 2023a; Wang et al., 2024; Meta, 2024; Team et al., 2024; Hurst et al., 2024; Anthropic, 2024). Early models like Flamingo, BLIP, and MiniGPT introduced a now-common architecture—comprising a vision encoder, a large language model (LLM), and a projection module to align modalities (Alayrac et al., 2022; Li et al., 2022; 2023b; Liu et al., 2023; Zhu et al., 2023)—but were limited by fixed-resolution inputs (typically 224×224), which led to information loss in high-resolution settings. Recent work addresses this by either adopting multi-crop strategies (Thapa et al., 2024; Guo et al., 2024; Liu et al., 2024) or enabling dynamic resolution processing through 2D Rotary Position Embeddings (RoPE) (Su

et al., 2024; Dehghani et al., 2023; Wang et al., 2024), allowing ViTs to natively handle varying image sizes. Large-scale instruction-tuning datasets such as LAION-5B (Schuhmann et al., 2022), ShareGPT4V (Chen et al., 2024), and LLaVA-Instruct-150K (Liu et al., 2023) have further enhanced reasoning capabilities across diverse domains. Additionally, token-efficient methods like the Perceiver Resampler (Jaegle et al., 2021; Alayrac et al., 2022) and spatial pooling techniques have enabled longer context lengths without incurring prohibitive compute.

VLMs have recently expanded into video understanding, introducing new challenges due to the high token count from sequential frames (Zohar et al., 2024; Wang et al., 2024; Lin et al., 2023; Team et al., 2024; Beyer et al., 2024). Strategies such as frame sampling and 3D spatiotemporal pooling have proven effective, aided by large-scale video datasets like WebVid-2M (Bain et al., 2021) and Video-ChatGPT (Maaz et al., 2023). Unified architectures such as OmniVLM and Gemini now demonstrate strong performance across both image and video tasks (Beyer et al., 2024; Hurst et al., 2024), underscoring the growing versatility of modern VLMs.

## B.2 Biomedical Vision-Language Models

Vision-Language Models (VLMs) have demonstrated strong potential in the biomedical domain, enabling tasks such as medical report generation, disease classification, and procedural understanding. Recent models like Dragonfly (Thapa et al., 2024), LLaVA-Med (Li et al., 2023a), Med-PaLM (Tu et al., 2024), and Med-Gemini (Saab et al., 2024) showcase the ability of VLMs to adapt to medical contexts. Their success is largely fueled by high-quality biomedical image-text datasets, including MedTrinity-25M (Xie et al., 2024), MIMIC-CXR (Johnson et al., 2019), CheXpert (Irvin et al., 2019), Quilt (Ikezogwo et al., 2023), OpenPath (Huang et al., 2023), and LLaVA-Med (Li et al., 2023a), which span modalities like X-rays, CTs, MRIs, and histopathology. These datasets provide radiology reports, disease annotations, and sometimes segmentation masks or bounding boxes—enabling diverse downstream tasks and improving domain-specific reasoning.

## B.3 Biomedical Video Datasets

Biomedical video analysis has gained momentum with the emergence of specialized datasets supporting various clinical applications and procedural understanding. Notable among these is Med-VidQA, which provides educational biomedical videos for video question answering and comprehension (Gupta et al., 2023), and NurViD, a video corpus of nursing procedures annotated at the action-step level (Hu et al., 2023). AVOS supports surgical scene analysis with 1,997 annotated surgical videos (Goodman et al., 2024), while MIMIC-IV-ECHO offers echocardiograms linked to clinical records, enabling research in cardiac function and diagnostic imaging (Gow et al., 2023). Cataract-1K contains annotated cataract surgery videos, facilitating ophthalmic phase recognition and skill assessment (Ghamsarian et al., 2024). Similarly, MedicalNarratives leverages educational medical videos to create a dataset focused on spatially grounding visual concepts, primarily targeting image-centric modalities such as CT, MRI, and X-rays (Ikezogwo et al., 2025). While their source material includes videos, the emphasis is on extracting and annotating static medical images rather than analyzing temporal dynamics or reasoning across multiple video frames. Together, these datasets have advanced the development of medical multimodal models.

Despite this progress, existing biomedical video datasets remain limited in both scale and scope, often targeting narrow tasks or specific modalities. MedVidQA and NurViD center on instructional and nursing procedures, AVOS targets open surgeries, and MIMIC-IV-ECHO is confined to echocardiography. Similarly, the EndoVis Challenge datasets emphasize surgical tool segmentation and phase recognition (Nwoye et al., 2023), offering limited support for broader multimodal learning. MedicalNarratives (Ikezogwo et al., 2025) also leverages educational medical videos but primarily focuses on extracting and annotating static medical images—such as CT, MRI, and X-rays—rather than enabling reasoning over dynamic video content. The lack of large-scale, diverse, and standardized datasets continues to hinder progress in biomedical video-language modeling. To address this gap, we introduce OpenBiomedVid, a large-scale and diverse corpus of educational biomedical videos sourced from YouTube, paired with carefully cleaned captions and an instruction-tuning dataset tailored for video-language modeling. Additionally, we release two standardized evaluation benchmarks—MIMICEchoQA and SurgeryVideoQA —designed to systematically assess biomedical video understanding.

# C   Additional Results

## C.1   Effect of Dataset Size on Performance.

To examine the impact of dataset size on performance, we evaluated the model across increasing proportions of the fine-tuning dataset and averaged the results for text, image, and video benchmarks. Supplementary Figure S1 illustrates the overall trends.

On image and video benchmarks, we observed a clear upward trajectory as more data was introduced, underscoring the importance of dataset scale for multimodal understanding. For image tasks, the `Qwen-2-VL-7B-biomed` model achieved most of its gains between 0% and 25% of the data, suggesting early saturation in image understanding. In contrast, video performance continued to improve more steadily up to 50% of the dataset, with further gains at a slower pace beyond that point. While the relatively small size of our fine-tuning dataset limits projections at larger scales, these trends suggest that both image and video performance could benefit from further data expansion.

On text benchmarks, however, the model exhibited minimal gains with additional data. The performance curve remained relatively flat across different dataset sizes, mirroring patterns observed earlier in Supplementary Figure 4. These results indicate that scaling biomedical video-caption data provides limited benefit for purely textual reasoning tasks. One likely reason is that the benchmark datasets—such as MedQA and PubMedQA—require specialized domain knowledge and structured question-answering capabilities, which may not be captured effectively through video-text instruction tuning.

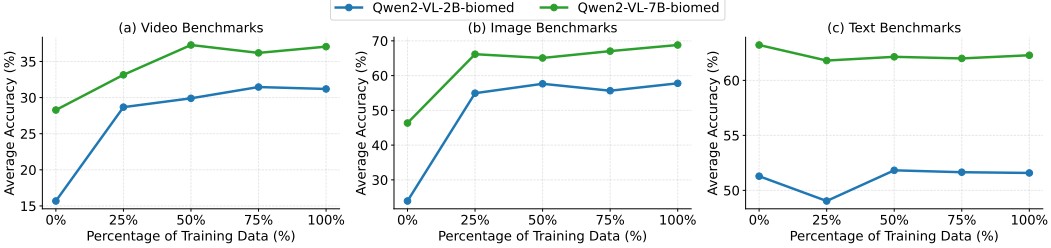

Figure S1: Average performance of the `Qwen2-VL-2B-biomed` and `Qwen2-VL-7B-biomed` models at various training checkpoints. The results highlight the impact of increasing training data, showing marginal gains for text benchmarks but substantial improvements for image and video benchmarks, underscoring the importance of multimodal data scaling.

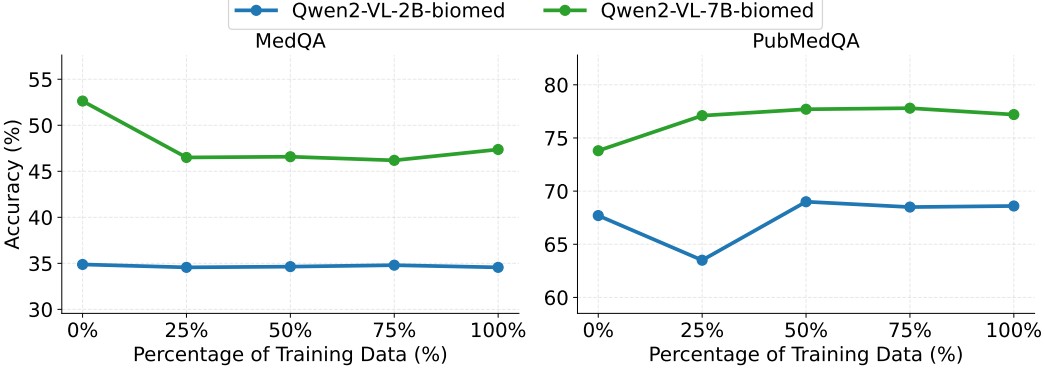

Figure S2: Performance of the `Qwen2-VL` models on text benchmarks at various training checkpoints. The figure illustrates how increasing the amount of training data impacts performance on text-based tasks, highlighting the marginal gains observed compared to image and video benchmarks.

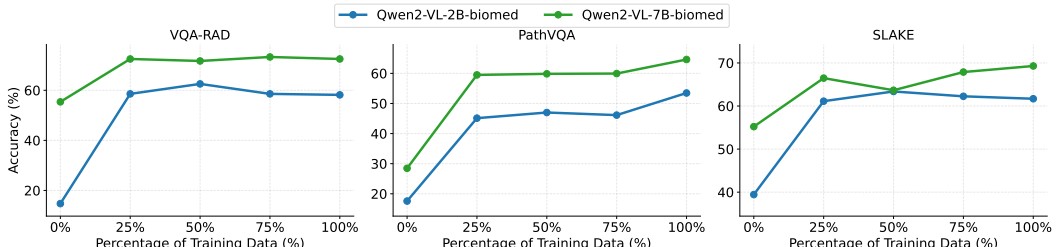

Figure S3: Performance of the `Qwen2-VL` models on image benchmarks at various training checkpoints. The figure illustrates how increasing the amount of training data leads to steady performance gains on image-based tasks, highlighting the importance of data scaling for enhancing visual understanding in biomedical contexts.

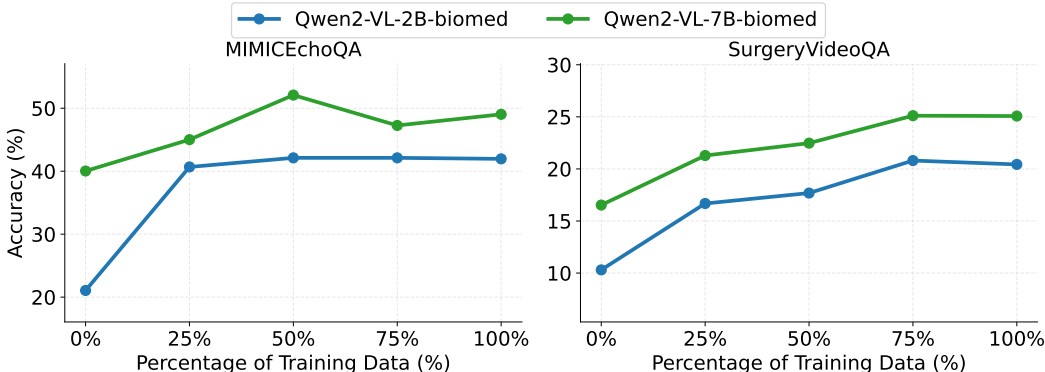

Figure S4: Performance of the `Qwen2-VL` models on video benchmarks at various training checkpoints. The figure shows a more substantial performance gain on MIMICEchoQA compared to SurgeryVideoQA, suggesting that the model benefits more significantly from additional training data in certain biomedical video contexts. The modest improvements on SurgeryVideoQA indicate potential challenges in capturing the complexity of surgical videos despite increased data.

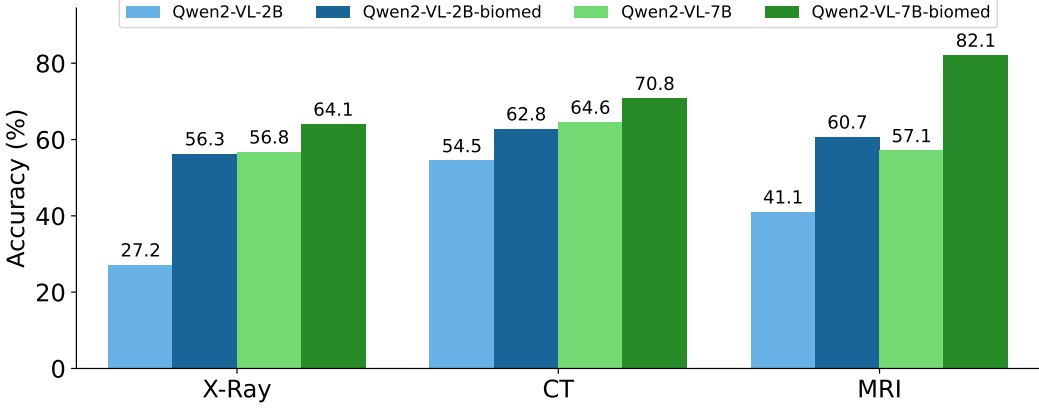

Figure S5: Performance of the `Qwen2-VL` models stratified by imaging modalities, including X-ray, CT scans, and MRI. The figure highlights how model accuracy varies across different modalities, with the highest performance observed on MRI images, followed closely by CT scans.

Table S1: Comparison of GPT-4o and Gemini-2.0-Flash judges across all models. Values are accuracy (%) with 95% bootstrap confidence intervals in parentheses. Agreement and Cohen's $\kappa$ measure inter-judge consistency.

| Model | GPT-4o (Judge) | Gemini-2.0-Flash (Judge) | Agreement (%) | Cohen $\kappa$ |
|---|---|---|---|---|
| Video-ChatGPT-7B | 7.9 (6.9–8.9) | 7.4 (6.4–8.5) | 95.8 | 0.70 |
| Video-LLaVA-7B | 9.2 (8.1–10.1) | 8.8 (7.7–9.8) | 95.0 | 0.69 |
| Phi-3.5-vision-instruct | 8.5 (7.5–9.5) | 8.1 (7.1–9.1) | 95.7 | 0.71 |
| Phi-4-multimodal-instruct | 8.5 (7.5–9.5) | 10.2 (9.2–11.4) | 94.6 | 0.68 |
| InternVideo2.5-Chat-8B | 12.2 (11.0–13.4) | 11.9 (10.7–13.2) | 94.4 | 0.74 |
| Qwen2-VL-7B-Instruct | 16.5 (14.9–18.3) | 13.2 (11.8–14.9) | 92.6 | 0.71 |
| Qwen2.5-VL-7B-Instruct | 17.9 (16.4–19.4) | 18.6 (17.2–20.1) | 91.9 | 0.73 |
| Qwen2-VL-72B-Instruct | 24.3 (21.9–26.5) | 26.8 (24.3–29.1) | 90.0 | 0.74 |
| Qwen2.5-VL-72B-Instruct | 24.3 (21.9–26.5) | 26.8 (24.3–29.1) | 90.0 | 0.74 |
| Gemini-2.0-Flash | 26.8 (23.0–30.6) | 27.2 (23.4–31.5) | 88.3 | 0.70 |
| GPT-4o | 35.8 (32.6–39.0) | 38.4 (35.2–42.6) | 90.0 | 0.78 |
| o4-mini | 46.6 (39.4–52.9) | 51.4 (44.7–58.2) | 86.5 | 0.73 |
| Qwen2-VL-2B-biomed | 20.4 (18.9–22.0) | 20.1 (18.6–21.7) | 93.2 | 0.79 |
| Qwen2-VL-7B-biomed | 25.1 (23.5–26.6) | 26.7 (25.1–28.3) | 91.3 | 0.77 |

Table S2: Performance of multimodal models before and after fine-tuning on OpenBiomedVid. Results show that OpenBiomedVid improves both InternVL3-8B and Qwen2-VL-72B across video and image-based biomedical QA benchmarks, confirming its architecture-agnostic utility.

| Task | InternVL3-8B | InternVL3-8B-Biomed | Qwen2-VL-72B | Qwen2-VL-72B-Biomed |
|---|---|---|---|---|
| MIMICEchoQA | 35.1 | 42.2 | 37.6 | 47.0 |
| SurgeryVideoQA | 15.5 | 20.1 | 25.3 | 28.1 |
| VQA-RAD | 67.3 | 72.1 | 77.0 | 76.5 |
| PathVQA | 58.0 | 65.8 | 73.8 | 71.0 |
| Slake | 64.0 | 66.2 | 75.1 | 75.0 |
| MedQA | 61.2 | 61.0 | 68.6 | 76.2 |
| PubMedQA | 62.6 | 67.5 | 76.3 | 79.4 |

## C.2 LLM-as-Judge Bias Analysis

To rigorously assess potential stylistic or evaluator bias introduced by GPT-4o, we conducted a comprehensive multi-judge analysis using both GPT-4o and Gemini-2.0-Flash across all 14 evaluated models. As shown in Supplementary Table S1, accuracy estimates under the two judges are highly consistent, with overlapping 95% CIs, 93.0% raw agreement, and Cohen's kappa = 0.748. Model rankings remain nearly identical (Spearman correlation = 0.972, Pearson r = 0.993).

To ensure evaluator independence, we also carried out a human study on 400 randomly sampled items across GPT-4o, Gemini-2.0-Flash, and our 2B/7B biomedical Qwen2-VL models, as shown in Supplementary Table S4. Both LLM judges show strong alignment with humans (90–95%, Cohen's kappa = 0.72–0.84 for GPT-4o; 89–95%, Cohen's kappa = 0.66–0.83 for Gemini), and also agree closely with each other (88–95%, Cohen's kappa = 0.65–0.85).

Finally, we highlight that GPT-4o is not generating captions de novo—the captions originate from the original YouTube audio; the LLM only cleans them and formulates Q/A pairs. Every stage of the pipeline includes human verification to prevent artifacts, and the entire SurgeryVideoQA and MIMICEchoQA benchmarks were fully reviewed by clinical experts.

## C.3 Evaluation Error Analysis

We conduct a comprehensive quantitative and qualitative error analysis of the model trained on our dataset (Qwen2-VL-7B-Biomed-Video) across both evaluation benchmarks. For MIMICEchoQA, we grouped errors by anatomical structure and ultrasound view and found that mistakes disproportionately concentrate in regions such as left atrium (68.4% error), pulmonary artery (66.7%), and right atrium (72.7%). Doppler and PSAX/PLAX great-vessel views also exhibit the highest error (e.g., DOPPLER:

Table S3: Comparison across Qwen2-VL 2B and 7B models (base, biomed-frame, biomed-video). Numbers are accuracy (%). Bold indicates the highest-performing model per task.

| Task | 2B-base | 2B-biomed-frame | 2B-biomed-video | 7B-base | 7B-biomed-frame | 7B-biomed-video |
|---|---|---|---|---|---|---|
| *Video Benchmarks* | | | | | | |
| MIMIC-EchoQA | 21.1 | 40.0 | **42.0** | 37.9 | 44.0 | **49.0** |
| SurgeryVideoQA | 10.3 | 15.6 | **20.4** | 16.5 | 23.0 | **25.1** |

Table S4: Human evaluation results on 100 sampled SurgeryVideoQA items. Accuracies are percentages judged correct. Agreement and Cohen's $\kappa$ quantify alignment between (H.) the human annotator, GPT-4o (GPT), and Gemini-2.0-Flash (Gem).

| Model | H. Acc | GPT Acc | Gem Acc | GPT↔H. | Gem↔H. | GPT↔Gem | $\kappa$(GPT,H.) | $\kappa$(Gem,H.) |
|---|---|---|---|---|---|---|---|---|
| GPT-4o | 23.0 | 27.0 | 29.0 | 90.0 | 94.0 | 94.0 | 0.73 | 0.84 |
| Gemini-2.0-Flash | 18.0 | 21.0 | 23.0 | 95.0 | 89.0 | 88.0 | 0.84 | 0.66 |
| Qwen2-VL-7B-biomed | 20.0 | 22.0 | 21.0 | 94.0 | 93.0 | 93.0 | 0.82 | 0.79 |
| Qwen2-VL-2B-biomed | 15.0 | 19.0 | 20.0 | 92.0 | 95.0 | 95.0 | 0.72 | 0.83 |

83%; PSAX great-vessels: 70–71%), consistent with the need for dynamic spectral reasoning. A manual review of 100 incorrect predictions further revealed clear, non-random patterns: roughly 60% of errors arise from severity miscalibration, where mild or severe findings are collapsed to moderate; 25–30% stem from threshold ambiguities (e.g., under-calling small pericardial effusions or over-calling physiologic aortic regurgitation); and 10% reflect systematic Ejection Fraction overestimation. These error modes likely arise from the inherent noise and subjectivity in report-derived labels (e.g., borderline severity categories, inter-expert disagreement, and variability in clinical phrasing).

For SurgeryVideoQA, we manually examined 100 randomly sampled incorrect predictions and again observed consistent, interpretable failure categories. Approximately 22% of errors involve anatomy/localization mistakes (e.g., confusing strap muscles with pectoralis major/minor, or pre-sacral space with the left iliac fossa). Instrument/technique/suture misidentification accounts for 25–33% of failures (e.g., predicting diathermy instead of a surgical knife). Procedural step or temporal confusion explains another 16%, where the model describes an earlier or later operative phase rather than the interval queried. Around 10% of errors arise from overly generic responses (e.g., "a knot" instead of "purse-string notch"), and 5–10% involve quantitative or extent misestimation (e.g., dissection distance, graft volume). Across all categories, most mistakes are near misses—the model generally identifies the correct procedure and high-level context but lacks the fine-grained, expert-level specificity required by the benchmark. These patterns are consistent with residual noise and label coarseness in the large, automatically curated YouTube-derived dataset.

## C.4 Training Data Quality Assessment

Human verification occurred at different depths for the training dataset versus the evaluation benchmarks. For the training dataset, human review was performed through random spot checks: a medical doctor inspected 1,000 sampled clips together with all associated Q/A pairs, covering a broad range of channels and content areas. This sampling strategy was intentional—the goal of the paper is to test whether models can learn from naturally noisy and heterogeneous educational videos, so we did not exhaustively clean the training corpus. In contrast, human verification for the evaluation benchmarks was conducted at the per-clip and per-QA level (i.e., full passes). MIMICEchoQA was fully curated by a board-certified cardiologist, and SurgeryVideoQA was fully curated by a general medical doctor; both experts removed any items showing common LLM failure modes such as generic answers, incorrect grounding, temporal mismatch, or hallucinated clinical content.

Regarding annotator counts, each benchmark involved two clinical annotators, and the frame-filtering stage involved two independent annotators, where we observed 95% human–model agreement and 93% human–human agreement on 1,000 frame-level samples immediately after training the SigLIP classifier. We agree that LLM-generated training captions/QAs may still contain occasional artifacts—this is inherent to the educational YouTube domain and aligned with the research question of whether models can learn effectively from such heterogeneous supervision.

To better quantify residual noise in the training corpus, we conducted an manual analysis of 100 randomly sampled caption–QA pairs, labeling each item with its primary issue when present. We found that 15% of samples contained ambiguous or low-information captions (e.g., generic openings such as "In this video..." that provide little clinical grounding), and 10% showed signs of incorrect grounding, where the caption was vague but the QA referred to specific anatomy or procedures. Only 5% exhibited clear LLM-style noisy phrasing (e.g., "the example under consideration..."). Importantly, 70% of samples were mostly normal and clinically grounded, consistent with our design choice to preserve some natural heterogeneity in the training data. As emphasized earlier, the two evaluation benchmarks—MIMICEchoQA and SurgeryVideoQA—are fully expert-curated and do not contain these artifacts. We will include representative cases in an error-analysis appendix to further document these patterns.

# D  Dataset Details

## D.1  Dataset Statistics

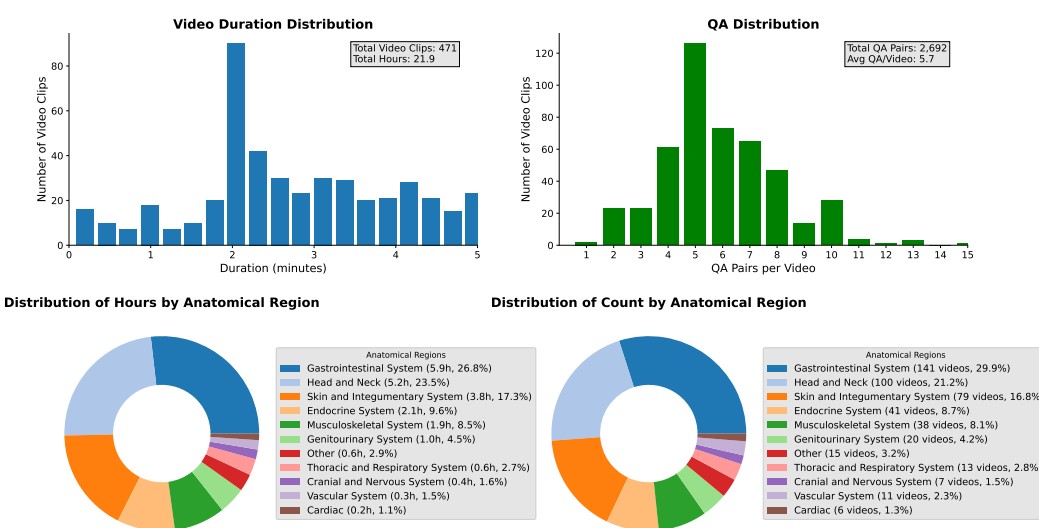

Figure S6: Statistics of the SurgeryVideoQA evaluation dataset, showing the distribution of videos and Q/A pairs across different anatomical regions and surgical procedures.

# E  Training Details

## E.1  Model Training

**VLM Training**    Traditional training of vision-language models (VLMs) involves padding sequences to a fixed maximum length, which becomes increasingly inefficient as dataset size and sequence variability grow. To improve training efficiency, we implement a sequence packing algorithm (Krell et al., 2022; Ding et al., 2024) tailored for VLMs. During preprocessing, we pre-allocate placeholders for visual input tokens and apply a best-fit decreasing algorithm to efficiently pack sequences of varying lengths within the maximum sequence limit.

To ensure attention is restricted to individual sequences within each packed batch, we utilize FlashAttention-2 (Dao, 2024), allowing each sequence to attend only to itself. During the forward pass, placeholders are dynamically replaced with embeddings from the vision encoder, avoiding redundant computation and significantly accelerating training. Detailed training hyperparameters are provided in Table S5. The total training time for `Qwen-2-VL-2B-Biomed` was 4.5 hours on a single node, while `Qwen-2-VL-7B-Biomed` required 5 hours on a single node.

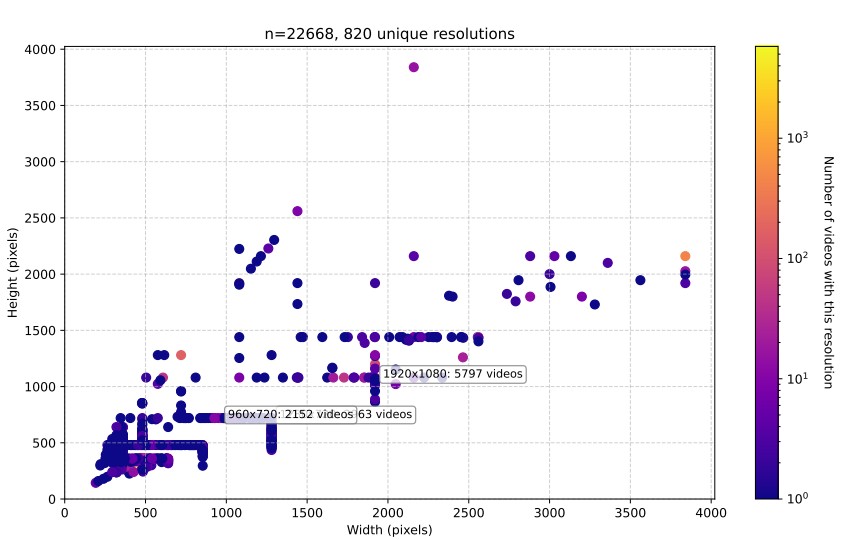

Figure S7: Video resolution distribution in the fine-tuning dataset

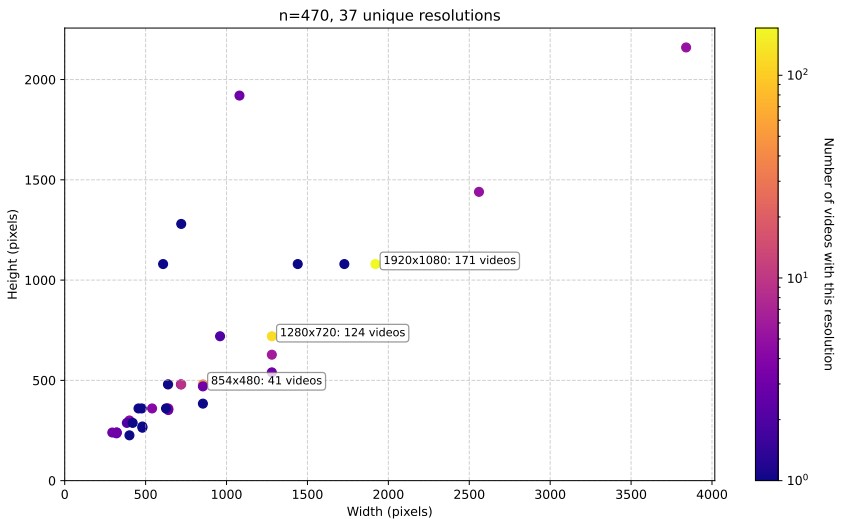

Figure S8: Video resolution distribution in the SurgeryVideoQA dataset.

Table S5: Training configuration for fine-tuning `Qwen-2-VL` models.

| Parameter | Qwen2-VL-2B | Qwen2-VL-7B |
|---|---|---|
| Max Sequence Length | 16,384 tokens | |
| Learning Rate | $1 \times 10^{-5}$ | |
| Optimizer | AdamW | |
| Gradient Accumulation Steps | 8 | |
| Batch Size (Per Device) | 1 | |
| Epochs | 1 | |
| Warmup Ratio | 0.1 | |
| Weight Decay | 0.01 | |
| Gradient Checkpointing | Enabled | |
| FP16/BF16 Precision | BF16 | |
| Learning Rate Scheduler | Cosine Decay | |
| GPUs Used | $8\times$ NVIDIA H100 | |

The original Qwen2-VL models sample videos at 2 frames per second (fps) and process them using 3D convolutions with a depth of two. They dynamically adjust frame resolution to constrain the total number of visual tokens per video to 16,384. In our fine-tuning, we retained most of these original settings but introduced a key modification: we sampled frames at 1 fps instead of 2 fps. This allowed us to use higher-resolution frames while maintaining the same token budget per video, thereby reducing the need for aggressive frame downsampling.

Table S6: Combined Training Configuration for Fine-tuning Siglip classifier.

| Parameter | Value |
|---|---|
| Learning Rate | $2 \times 10^{-5}$ |
| Batch Size (Per Device) | 32 |
| Epochs | 1 |
| Weight Decay | 0.01 |
| GPUs Used | $8\times$ NVIDIA H100 |

**Classifier Training** For classifier training, we trained the linear layer on top of the SigLIP vision encoder (Zhai et al., 2023). The total training time was less than 1 hour. The training parameter is shown in Table S6.

# F   Prompt Details

## F.1   Medical Image Classification

---

**Prompt for Medical Image Classification**

You are an expert at identifying biomedical content in images. Your task is to classify each image as either True or False based on the following structured criteria:

**Classification Criteria:**
1. True: The image predominantly features biomedical visuals. Examples of valid biomedical visuals include, but are not limited to:
    - Echocardiograms
    - Sonograms
    - Surgical procedures
    - Animations illustrating biological processes
    - Ultrasounds
    - X-rays
    - Body organs or tissues
    - CT scans
    - Cells or cellular structures
    - Endoscopy
    - Microscopic images
    - Fluoroscopy
    - Fundoscope
    - PET scans
    - Otoscopy
    - Mammography
    - Dermatoscopy
    - Cystoscopy
    - Angiogram
    - Colonoscopy
    - MRI
    - Arthroscopy

The biomedical visuals must be the dominant part of the image.

Additional context:
    - Biomedical text, even if descriptive of biomedical content, does not qualify.
    - The presence of medical professionals alone, without dominant biomedical visuals, does not qualify.

2. False: Any image that does not meet the above "True" criteria. Examples include:
    - Images where biomedical content is not the dominant feature.
    - Images with only biomedical text.
    - Images with only medical professionals without accompanying biomedical visuals.

---

## F.2 Filter and Clean Captions

**Prompt for Filtering and Cleaning Captions**

You are an AI assistant that determines if a caption describes relevant biomedical video content and cleans it to be neutral and descriptive.

**STEP 1: FILTER FOR BIOMEDICAL RELEVANCE**
The caption must meet BOTH criteria to be considered biomedical:

**1. Describes observable visual content:**
- Medical imaging (ultrasounds, X-rays, MRI, CT scans, etc.)
- Clinical procedures or examinations
- Surgical operations
- Microscopic views
- Anatomical demonstrations
- Medical device operations
- Patient assessments
- Laboratory procedures with visual components

**2. Contains specific biomedical elements:**
- Not just medical settings or personnel
- Must describe actual medical/biological content
- Should reference visual elements that would be seen in the video

**STEP 2: CLEAN THE CAPTION**
Clean all captions following these guidelines:

**1. NO HALLUCINATION:**
- Do not make up information
- All cleaned captions must be entirely grounded in the original

**2. Standardized Terminology:**
- Use consistent and precise terminology
- Replace colloquial terms with standardized phrases

**3. Level of Detail:**
- Reflect only explicitly stated details
- Do not infer additional information

**4. Spatial and Temporal Context:**
- Include spatial/temporal references only when explicitly described
- Avoid assuming spatial or temporal details

**5. Remove Noise but Keep Context:**
- Remove non-visual information unless essential

**6. Neutral and Objective Language:**
- Remove conversational tone and subjective comments

**Output Format:**
`<is_biomedical>`: true/false
`<cleaned_caption>`: cleaned caption

## F.3   Generate Question Answer Pairs

---

**Prompt for Generating Question Answer Pairs**

You are an expert in biomedical video analysis and medical diagnosis. Your task is to generate high-quality question-answer (Q/A) pairs for a biomedical video benchmark dataset.

**Generate only Q/A pairs that meet all of these criteria:**

1.  The answer must require watching the video. It should be based on visual details observed in the video, not general medical knowledge.

2. Focus on specific biomedical aspects:
   - Diagnoses, such as identifying visible pathologies
   - Procedures, including surgical techniques or medical interventions
   - Medical findings, such as abnormalities in tissues or organs
   - Anatomical features, including structural changes in biological samples

3.  Avoid generic, basic, or trivial questions. Ensure they are medically relevant and non-obvious.

4.  The questions must be strictly based on the visual content of the video. Do not introduce hallucinated or unrelated information.

5.  Ensure that each question focuses on a single, specific detail. Avoid compound or multi-part questions.

6. Use diverse question formats, such as:
   - What is the ejection fraction of the heart as shown in the video?
   - What abnormalities are detected in the lungs as shown in the video?
   - Where is the ulcer located?
   - Which visual signs indicate arterial plaque in the coronary artery?
   - When does peristalsis slow down in the small intestine?
   - Which feature distinguishes this tumor from surrounding tissue?

7. Keep answers concise and direct (few words), avoiding unnecessary elaboration.

Return the generated Q/A pairs as a valid JSON array in this exact format:
{"question": "question text", "answer": "answer text"}
{"question": "question text", "answer": "answer text"}

---

## F.4 Generate Metadata

---

**Prompt for Video Metadata Generation**

Based on this medical video caption: {cleaned_caption}

**Select exactly ONE video modality from:**
- Echocardiography
- Ultrasound
- CT Imaging
- MRI
- Angiography
- X-ray
- Fluoroscopy
- Endoscopy and Laparoscopic Surgery
- Surgical and Procedural
- Anatomy and Biological Processes
- Microscopy
- Other

**And exactly ONE anatomical category from:**
- Cardiac
- Vascular System
- Musculoskeletal System
- Cranial and Nervous System
- Thoracic and Respiratory System
- Gastrointestinal System
- Genitourinary System
- Head and Neck
- Endocrine System
- Skin and Integumentary System
- Other

**Output Format:**
`<modality>`: Single most relevant video modality (no commas)
`<anatomical_region>`: Single most relevant anatomical category (no commas)

Base your selection purely on the primary focus described in the caption. Choose the single most central category that best matches the main content. Do not include multiple categories or variations.

You must select exactly one option from each list above - do not create new categories or combine existing ones.

---

## F.5    Generate SurgeryVideoQA Benchmark

---

**Prompt for Creating Surgical Benchmark**

You are an expert in surgical video analysis and intraoperative decision-making. Your task is to generate high-quality question-answer (Q/A) pairs for an open-surgery video benchmark dataset.

**Generate only Q/A pairs that meet all of these criteria:**

**1. The answer must require watching the video.**
   - The answer should be based on visual details observed in the video, not general surgical knowledge.
   - Avoid questions that could be answered without analyzing the surgical footage.

**2. The questions must be strictly grounded in the video caption.**
   - Do not introduce hallucinated or unrelated information.
   - Ensure that each question is fully supported by the caption.

**3. The questions must focus on these critical open-surgery aspects:**
   - Surgical procedures (e.g., dissection, hemostasis, suturing, anastomosis)
   - Anatomical structures (e.g., vessels, organs, tissues, nerves)
   - Pathological findings (e.g., necrosis, tumor, hemorrhage, infection)
   - Surgical instruments (e.g., scalpel, forceps, electrocautery, retractors)
   - Complications (e.g., excessive bleeding, ischemia, perforation)
   - Intraoperative decision-making (e.g., changing instruments, modifying the approach)

**4. The questions must be direct, precise, and non-trivial.**
   - Avoid vague or general questions (e.g., "What is happening?" is not acceptable).
   - Each question must target a single, specific surgical detail.
   - Do not ask broad, multi-part, or ambiguous questions.

**5. The answers must be concise (only a few words).**
   - No explanations or unnecessary elaboration.
   - Keep answers strictly factual and directly grounded in the caption.

Return the generated Q/A pairs as a valid JSON array in this exact format:
{"question": "question text", "answer": "answer text"}
{"question": "question text", "answer": "answer text"}

---

## F.6 Generate MIMICEchoQA Benchmark

---

**Prompt for Creating Echocardiogram Benchmark**

You are an expert in cardiology and echocardiogram interpretation. Your task is to generate high-quality question-answer (Q/A) pairs based strictly on the given **echocardiogram video caption**.

**Generate only Q/A pairs that meet all of these criteria:**

**1. Only focus on heart, valve, and chamber abnormalities and ejection fraction.**
   - The questions must target visible findings such as valve regurgitation, stenosis, chamber dilation, wall motion abnormalities, or ejection fraction.
   - Avoid questions that could be answered without analyzing the echocardiogram footage.

**2. Do not use any external information beyond what is explicitly in the caption.**
   - Do not introduce hallucinated or unrelated information.
   - Ensure that each question is fully supported by the caption.

**3. Frame questions as if they are about the echocardiogram video itself, not a text report.**
   - Questions should refer directly to observations in the video, not inferred or historical data.

**4. Do not hallucinate findings—if the caption does not provide enough information, return None.**
   - If the caption lacks specific details, avoid making assumptions or introducing unsupported information.

**5. Ensure answers are short and direct.**
   - No explanations or unnecessary elaboration.
   - Keep answers strictly factual and directly grounded in the caption.

**6. Only include findings that are currently visible in the echocardiogram.**
   - Do not reference past medical history or prior diagnoses.
   - Focus solely on present and visible findings in the video.

**7. Return the generated Q/A pairs as a valid JSON array in this exact format:**
{"question": "question text", "answer": "answer text"}

---

## F.7    Video Evaluation

---

**Prompt for Video Evaluation**

You are an expert evaluator for medical image and video understanding. Your task is to compare a gold standard answer to a predicted answer and determine the similarity score.

Ignore minor differences in formatting and verbosity.  Focus on whether the predicted answer conveys the same essential meaning as the gold answer.

Instructions:
Assign a score of 0 or 1 based on how similar the prediction is to the gold answer:
1: Correct - The prediction is mostly correct and captures the essential meaning, with minor errors being tolerable.
0: Incorrect - The prediction is largely incorrect or unrelated.

You must respond with ONLY a single integer number: 0 or 1.

Question: {question}
Gold Answer: {gold_answer}
Predicted Answer: {pred_answer}

---

# G  Qualitative Examples

## G.1  Finetuning Dataset

---

**Example #1**

**Modality:** Ultrasound
**Anatomical Region:** Other
**Query:** ultrasound
**Link:** https://youtu.be/akxn2mI2rQc

**Cleaned Caption:** Breast ultrasounds are utilized to assess the nature of breast masses. A mass containing only fluid indicates a benign simple cyst, which typically does not necessitate further evaluation. In contrast, solid masses have the potential to be breast cancer. The procedure for a breast ultrasound requires no special preparation. During the examination, a transducer is used to glide over the skin, capturing images of the internal structures of the breast efficiently and painlessly.

**Question:** What characteristic observed in the breast mass indicates it is a benign simple cyst as shown in the video?
**Answer:** The mass contains only fluid.

**Question:** What is the procedure used during the breast ultrasound in the video?
**Answer:** Gliding a transducer over the skin.

---

**Example #2**

**Modality:** MRI
**Anatomical Region:** Genitourinary System
**Query:** dynamic_mri
**Link:** https://youtu.be/251EKL3SKao

**Cleaned Caption:** The presentation begins by illustrating the progression of a breast lesion. Initially, the lesion showed an increase in intensity, eventually washing out, and subsequently opened up, indicating a malignant breast lesion rather than a benign one. This case serves as an example.

The second case concerns a patient undergoing neoadjuvant chemotherapy. Post-chemotherapy ultrasound indicated that the mass size had increased compared to the pre-chemotherapy size, raising concerns about the treatment's effectiveness. To address this issue, a breast MRI was conducted to evaluate the size and activity of the tumor following chemotherapy.

The T1-weighted fat-saturated images before contrast injection showed areas of increased signal intensity. To further assess the condition, a dynamic breast MRI was performed after contrast administration. This technique involves acquiring images both before and after contrast injection, allowing for the identification of any post-contrast enhancement areas. Post-contrast subtraction images were obtained, with contrast being applied six times, though only five sequences are demonstrated here.

Initially, no enhancement was observed during the first sequence. In the third stage, areas of enhancement began to appear. These included slight central necrosis, indicating suspicious and potentially malignant characteristics. The progression showed a distinct pattern in the lymph nodes, where their visibility improved, becoming more evident over subsequent sequences.

**Question:** What does the initial increase in intensity and subsequent washout of the lesion in the breast suggest?
**Answer:** A malignant breast lesion.

**Question:** What was the unexpected finding in the ultrasound after neoadjuvant chemotherapy?
**Answer:** The mass size increased.

**Question:** What was observed in the T1-weighted fat-saturated images before contrast injection?
**Answer:** Increased signal intensity areas.

**Question:** What changes were visible in the lymph nodes during the post-contrast sequences?
**Answer:** Improved visibility.

## Example #3

**Modality:** Echocardiography
**Anatomical Region:** Cardiac
**Query:** echocardiogram
**Link:** https://youtu.be/dic24ssb7Mk

5:21                                                                                              7:03

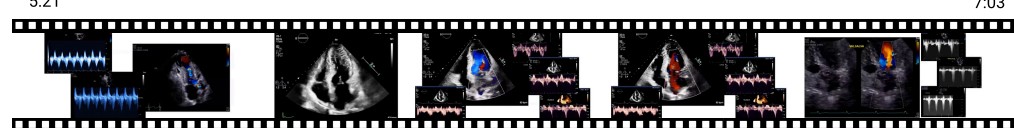

**Cleaned Caption:** During early diastole, a notable gradient is observed. Intercavitary gradients are commonly detected in patients exhibiting concentric hypertrophy, large or malformed papillary muscles, or hyperdynamic cardiac function. The patient in question demonstrates moderate concentric hypertrophy and a reduced left ventricular cavity size. To ascertain the precise location of the gradient, color Doppler imaging is employed, revealing turbulence throughout the left ventricular cavity. Pulse wave Doppler is also utilized to pinpoint the obstruction. In the initial pulse wave Doppler signal, the sonographer positions it within the left ventricular outflow tract (LVOT), where laminar flow appears, indicating a normal LVOT waveform. When the sample volume is adjusted to the mid-left ventricle, a late peaking waveform emerges, identifying the region of turbulence. Continuous wave Doppler analysis highlights the late peaking gradient, distinguished from the aortic gradient by its triangular shape and lower velocity. This patient may benefit from a provocative maneuver to evaluate the presence of a provokable gradient. Despite being challenging, color Doppler imaging can be conducted during the Valsalva maneuver to observe changes in LVOT turbulence and mitral regurgitation under provocation. The sonographer acquires an extended loop capturing images before the patient initiates the Valsalva maneuver and during the Valsalva strain phase, during which worsening turbulence in the LVOT and mitral regurgitation is observed.

**Question:** What feature differentiates the late peaking gradient seen in the video from the aortic gradient?
**Answer:** Triangular shape and lower velocity.

**Question:** What change occurs in the LVOT and mitral regurgitation during the Valsalva maneuver as observed in the video?
**Answer:** Increased turbulence and worsened mitral regurgitation.

**Question:** Where is the initial pulse wave Doppler signal placed, as shown in the video, to observe normal laminar flow?
**Answer:** Left ventricular outflow tract (LVOT).

**Question:** What is revealed when the sample volume is placed in the mid-left ventricle as shown in the video?
**Answer:** Late peaking waveform indicating turbulence.

**Example #4**

**Modality:** Angiography
**Anatomical Region:** Vascular System
**Query:** angiogram
**Link:** https://youtu.be/NKTVKRMfONU

6:06                                                                                          8:44

**Cleaned Caption:** The examination of the posterior circulation adopts an approach akin to that used for assessing the anterior circulation. Initially focusing on the right side, the vertebral artery is traced as it ascends, subsequently turning posteriorly to make another turn before entering the dura. It is common for vertebral arteries to exhibit slight narrowing at this juncture. The intracranial portion then ascends to join the contralateral vertebral artery. Although the posterior inferior cerebellar artery (PICA) was not visible on this side, a clear view is anticipated on the opposite side.

Upon observing the vertebral artery, which demonstrates some tortuosity by appearing intermittently through the slices, it is seen entering the foramen magnum and re-entering the dura. As the view progresses upwards, a small PICA becomes apparent, supplying the inferior region of the left cerebellum, with additional swirling PICA branches visible. The vertebral artery proceeds to merge with the right side, forming the basilar artery.

Attention is drawn to the basilar artery's first significant branch, the anterior inferior cerebellar artery (AICA). It is identified on the right, near the level of the internal auditory canal, which correlates with the vessel's proximity to the ICA. A slightly smaller AICA is noted on the opposite side, correlating with the presence of a larger PICA on the same side, indicating potential variability in vascular supply.

The superior cerebellar artery, appearing right before the termination of the basilar artery, is observed on the right, with a paired counterpart on the left encircling the midbrain. These arteries show symmetry without narrowing. As the view reaches the basilar artery's tip, the midbrain is examined for aneurysms. The posterior cerebral artery (PCA) bifurcation is visible, with the right and left PCA traced forward. Although a posterior communicating artery (PCOM) is not clearly visible, the PCA branches are seen around the midbrain, extending along the tentorium and supplying the occipital lobe. Similar features are represented on the left side, with the PCA circling the cerebral peduncle along the tentorium to supply the inferior occipital lobe. Both sides display symmetry, with no evidence of occlusion.

**Question:** Which artery connects with the basilar artery on the left side as shown in the video?
**Answer:** Vertebral artery.

**Question:** What structural feature is observed in the superior cerebellar arteries in the video?
**Answer:** Symmetry without narrowing.

**Question:** Which artery branches near the internal auditory canal as it is shown on the right side in the video?
**Answer:** Anterior inferior cerebellar artery (AICA).

**Question:** What is the visibility status of the posterior inferior cerebellar artery (PICA) on the right side as shown in the video?
**Answer:** It is not visible.

**Example #5**

**Modality:** Microscopy
**Anatomical Region:** Other
**Query:** microscopy
**Link:** https://youtu.be/Iv7eGCPVaAk

1:16                                                                          2:50

**Cleaned Caption:** The process begins with centering the slide on the microscope stage, which is then turned on for observation. As the slide is examined, the cell wall, identified as the dark outer layer, comes into view. The purple area represents the cytoplasm, within which are visible small circular structures, identified as nuclei inside the cell.
To demonstrate plasmolysis, a salt water solution is prepared. The salt is applied to the right side of the slide, while a paper towel is placed on the left side. This setup draws the salt water across the slide, effectively removing the water added earlier. In a time-lapsed sequence, the dramatic impact of the salt water is observed on the cells. The osmotic effect causes the water to be drawn out of the cells, resulting in a noticeable shrinkage of the cell interiors, while the cell wall remains unchanged in size. This shrinkage of the onion cells is identified as plasmolysis.
Next, the salt water is removed and replaced with fresh water using the same technique. Observation reveals that the cells swell back to their original size, a demonstration of osmosis. This process of alternating shrinkage and swelling can be repeated multiple times, illustrating osmosis in action within the cells. The experiment concludes with a clarification of the observed phenomena.

**Question:** What cellular process is demonstrated when the salt water causes shrinkage of the cell interiors while keeping the cell wall unchanged in size?
**Answer:** Plasmolysis.

**Question:** What happens to the cells when the salt water is removed and replaced with fresh water?
**Answer:** The cells swell back to their original size.

## Example #6

**Modality:** Endoscopy and Laparoscopic Surgery
**Anatomical Region:** Gastrointestinal System
**Query:** endoscopy
**Link:** https://youtu.be/qZjoBAGjlFo

0:26                                                                                       5:15

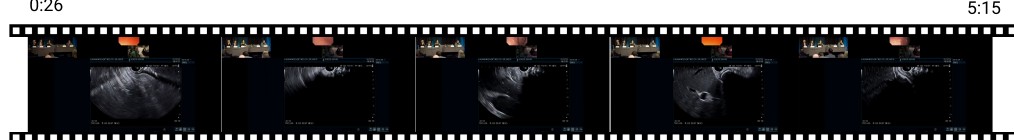

**Cleaned Caption:** The video begins on the morning of day two, highlighting the importance of the case being presented, which primarily focuses on organ preservation, specifically that of the gallbladder. Despite the presence of stones causing symptoms in the patient, the gallbladder is otherwise healthy. The case description underscores that the use of LAMS, a device not approved for gallbladder drainage in the U.S., aligns with standard care due to the patient's unsuitability for surgery. The patient was assessed by interventional radiology (IR), which concluded that internal drainage is preferable over external drainage for a long-term solution. Should surgery become viable, a percutaneous cholecystostomy would likely have been performed. The intent of using LAMS here, although off-label, is to preserve the gallbladder by enabling internal drainage, consistent with the original aim of the technology.
The video proceeds with several teaching points, notably the use of a therapeutic echo-endoscope with a large 3.7 mm channel necessary for placing the LAMS. It highlights the preference for a transduodenal method for gallbladder drainage, as opposed to the transantral approach, to reduce the risk of food entering the gallbladder through the LAMS. Ensuring clear avoidance of the pylorus with the LAMS flange is stressed, necessitating a thorough endoscopic evaluation to confirm the bulb is normal and free from any pathology. The video guides the viewer through positioning the LAMS correctly, avoiding the apex and positioning near, but not at, the pylorus due to the ultrasound's leading viewpoint relative to the endoscopic one.
During the procedure, the transition to ultrasound imaging is made to visualize the gallbladder and identify complications such as a significant stone occupying most of the gallbladder space, leaving minimal bile. This necessitates filling the gallbladder with saline via an FNA needle already attached to the scope, expanding it to provide a large enough target for the LAMS placement. The video also addresses the challenge presented by the stone absorbing the cautery energy, complicating penetration efforts. The solution proposed involves using additional saline to facilitate the procedure, paralleling aspects of the EDGE procedure discussed in the video.

**Question:** What device is used in the video for gallbladder drainage, even though it's not approved for this use in the U.S.?
**Answer:** LAMS.

**Question:** Which method is preferred for gallbladder drainage to minimize risk in the video?
**Answer:** Transduodenal approach.

**Question:** What is the purpose of filling the gallbladder with saline in the video?
**Answer:** To create a larger target for LAMS placement.

**Question:** What challenge is presented by the gallbladder stone during the procedure?
**Answer:** Absorbing cautery energy.

## Example #7

**Modality:** Anatomy and Biological Processes
**Anatomical Region:** Musculoskeletal System
**Query:** ct_scan
**Link:** https://youtu.be/fUZx47OkOvs

0:24                                                                                    5:12

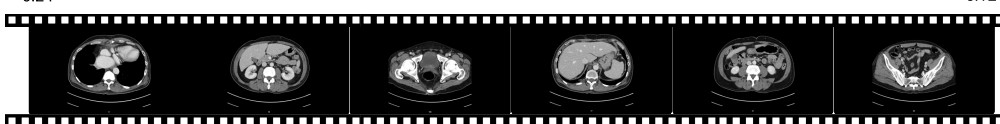

**Cleaned Caption:** The video initiates by examining the abdominal wall, focusing on Camper's fascia, identified as the dark tissue, which is essentially fat. The video shows how the skin and Camper's fascia behave intriguingly near the umbilicus, indicating the connection between the umbilicus and the liver. This connection comprises the round ligament, part of the falciform ligament, which separates the liver's right and left lobes, extending up to the umbilicus.
Attention then shifts to the vertebral column starting with the thoracic vertebrae, identified by their articulation with the ribs. Moving downward, the transition to lumbar vertebrae is noted due to the absence of ribs and the presence of prominent mammillary processes. Further down, the video identifies the sacrum and the sacroiliac joint—a synovial joint with limited mobility. The focus shifts back to vertebral features, highlighting the spinous and transverse processes, lamina, vertebral canal, spongy vertebral body, and pedicles. As the examination moves, the pedicle disappearance signals an intervertebral foramen. This part of the video then highlights the formation of the os coxa by three bones: the ilium, ischium, and pubis.
Exploration continues with the femur's appearance, particularly its head and greater trochanter, forming a ball-and-socket joint at the hip with the acetabulum. Observation highlights the pubis, forming the pubic symphysis, and the ischium with the ischial tuberosity.
Attention shifts to the abdominal wall muscles, showing three distinct layers: external oblique, internal oblique, and transverse abdominis, with the rectus abdominis in the front and the linea alba connecting them. The diaphragm is seen next, similar in tone to the liver, distinguishable by following the muscle to the right and left crura.
Focus then moves to the psoas major muscle, originating from the lumbar vertebrae and enlarging distally, with the quadratus lumborum muscle originating from transverse processes. The iliacus is then located within the iliac fossa, forming the iliopsoas with the psoas major under the inguinal ligament, serving as major hip flexors.
The narrative concludes with a transition to the cardiovascular system.

**Question:** What anatomical feature is represented by the dark tissue identified in the video?
**Answer:** Camper's fascia.

**Question:** Which ligament extends from the umbilicus to the liver as shown in the video?
**Answer:** Round ligament.

**Question:** Which characteristic differentiates lumbar vertebrae from thoracic vertebrae in the video?
**Answer:** Presence of mammillary processes.

**Question:** In the video, what forms the ball-and-socket joint at the hip?
**Answer:** The femur head and the acetabulum.

**Question:** What structure is identified when the pedicles disappear in the video?
**Answer:** Intervertebral foramen.

## Example #8

**Modality:** CT Imaging
**Anatomical Region:** Gastrointestinal System
**Query:** ct_scan
**Link:** https://youtu.be/tlaS22bIiVE

3:20      6:40

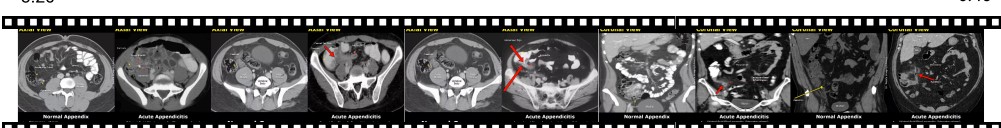

**Cleaned Caption:** The appendix is identified as the hyperdense bright structure, known as the appendiculite. The appendiculite always appears bright due to its composition of calcified fecal material. The presence of an appendiculite can facilitate the location of the appendix. In certain cases of appendicitis, a CECO bar sign may be observed. This sign involves inflamed tissue seen between the cecum, which is filled with contrast, and the inflamed appendix. The CECO bar sign is only visible when contrast medium is present in the cecum. Another observable sign in a contrast-filled cecum is the arrowhead sign, characterized by the arrowhead shape of contrast at the appendiceal orifice, which can appear in some appendicitis cases. The coronal blade provides a valuable perspective for viewing the inflamed appendix. In a normal image, the appendix may not be visible, which is a common occurrence due to various factors such as the appendix's variable location, bowel gases, and fecal matter that can obscure the normal appendix. Conversely, an inflamed appendix is more easily identifiable. An enlarged and swollen appendix is seen here. Administration of IV contrast enhances the walls of the appendix, revealing peri-appendiceal inflammation. The density and heterogeneity of the fat around the appendix indicate inflammation, appearing brighter than expected. Another image of the same case shows the enlarged appendix with peri-appendiceal inflammation more clearly. Another coronal image shows the normal appendix filled with gas, identified by the black area within. The presence of gas is normal since the appendix is not enlarged. Conversely, an image of appendicitis displays peri-appendiceal inflammation, with bright high-density areas in the surrounding fat. The appendix appears enlarged and gas-filled, with intraluminal gas indicating possible gangrenous appendicitis. This type of inflammation is of greater concern due to its severity.

**Question:** What visual indication identifies the presence of an appendiculite in the video?
**Answer:** It appears as a hyperdense bright structure due to calcified material.

**Question:** What sign can be seen when a contrast medium is present in the cecum during appendicitis?
**Answer:** Arrowhead sign.

**Question:** What feature indicates peri-appendiceal inflammation in the video?
**Answer:** Bright high-density areas in the surrounding fat.

**Question:** How is a normal appendix identified in the coronal image shown in the video?
**Answer:** By the presence of a dark area signifying gas filling.

## Example #9

**Modality:** Fluoroscopy
**Anatomical Region:** Musculoskeletal System
**Query:** fluoroscopy
**Link:** https://youtu.be/NIL_Ttvl6Co

4:08                                                                                            8:36

**Cleaned Caption:** The only way for these facet joints to open or gap like this is through tearing of the capsular ligaments. The current view is of the right side of the patient's neck. She has torn the capsular ligament between the vertebrae C5, C6, C6, C7, and C7, T1, all on one side of her cervical spine. The examination will soon proceed to the opposite side. Just like all other projections, this process is repeated three times. In a moment, the patient will be turned to face the opposite direction to examine the left facet joints.
On viewing the completely opposite side, before she lowers her chin towards her chest, one can observe a significant gapping of the facet joints. The view will be recounted to ensure clarity of the segments being examined: C1, C2, C3, C4, C5, C6, C7, and then T1. Notice the marked gapping at C5 on C6, indicative of a torn capsular ligament. There is a substantial gap between T1 and T7, which becomes more apparent as the chin moves towards the chest. Observing the levels C5, C6, C7, and T1, there is noticeable separation. This visual represents the final projection of a cervical DMX study, specifically the A-P open mouth lateral bending.
This view involves the patient opening her mouth, attempting to bring the right ear to the right shoulder and the left ear to the left shoulder. C2 is identifiable with the presence of the pyramid and the two triangular bones, C1 sits atop this, supported by a bone ring. Upon initiating movement, C1 begins to misalign relative to C2, with the lateral mass on both sides descending further off the edge with each leaning motion.
A small arrow marks where C1 dislocates from the edge of C2, indicating potential injury or tearing of the opposite alar and accessory ligaments. The same motion will be repeated with the left ear towards the left shoulder. Though the freeze-frame timing is slightly late, C1 is observed to be even further off C2 on the left side, marked by a left indicator. Thus, this examination suggests injury to both right and left alar and accessory ligaments in this patient.

**Question:** Which specific ligament tear is visible on the right side of the patient's cervical spine in the video?
**Answer:** Torn capsular ligament between C5 and C6.

**Question:** What specific motion increases the visibility of the gap between the facet joints on the opposite side of the cervical spine?
**Answer:** The patient lowering her chin towards her chest.

**Question:** What anatomical change is noted when the patient leans her head in the video?
**Answer:** C1 descends further off the edge of C2 on both sides.

**Question:** Which ligament injury is indicated by the motion where C1 dislocates from C2?
**Answer:** Injury to the right and left alar and accessory ligaments.

### Example #10

**Modality:** X-ray
**Anatomical Region:** Head and Neck
**Query:** bone_density_scan
**Link:** https://youtu.be/RqfqQqtgADQ

2:24                                                                                    4:22

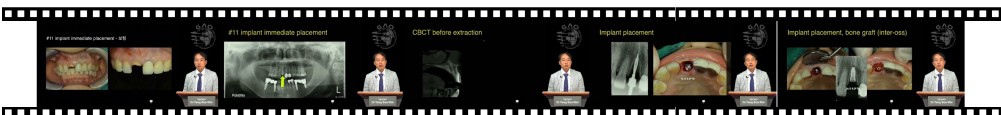

**Cleaned Caption:** The patient, who was over 65 years old, was eligible for insurance coverage for implants. In addition to this, the post-cork crown option was explained. However, the patient expressed a preference for the implant procedure. Consequently, an implant was placed following the extraction. The accompanying photo includes a panoramic X-ray taken prior to the extraction, as well as a clinical photo of the mandibular occlusal surface. Results from the CBCT performed before the extraction showed that, although the buccal bone was thin, it was healthy. This assessment allowed for the determination that immediate placement was viable, facilitating the establishment of a treatment plan.
The treatment stages are as follows: To ensure a safe extraction, the lingual side of the tooth was cut, and the tooth was separated in the mesiodistal direction. The extraction process was then carefully executed, followed by a periapical X-ray to confirm the implant path. Given the depth of the extraction site, a 2.2 mm drill was used in place of a standard guide pin to verify parallelism with adjacent teeth. Subsequently, the implant was placed while ensuring a buccal side gap of at least 2 mm. Once the implant was in position, the adjacent teeth and surrounding conditions were assessed to confirm the placement depth, verified through another periapical X-ray. Upon securing the appropriate depth, bone grafting was performed using Allograft to fill the space between the buccal bone and the implant.

**Question:** Which diagnostic imaging was performed before the extraction to assess buccal bone condition?
**Answer:** CBCT scan.

**Question:** What modification was made to the tooth before extraction according to the video?
**Answer:** Tooth was cut on the lingual side.

**Question:** What size drill was used to verify parallelism with adjacent teeth during the implant procedure?
**Answer:** 2.2 mm drill.

**Question:** What was used for bone grafting between the buccal bone and the implant?
**Answer:** Allograft.

## Example #11

**Modality:** Other
**Anatomical Region:** Skin and Integumentary System
**Query:** dermatoscopy
**Link:** https://youtu.be/c0jsOLOMwcY

4:39                                                    5:42

**Cleaned Caption:** The video presents an examination of an acral nevus. In one case, no changes were observed in the acral nevus over a period of four months. However, in another instance, alterations were detected in a melanocytic lesion on the upper arm of a man in his 60s after four months of follow-up. The subsequent image reveals fibrosis and white areas at the 10 o'clock position, along with the emergence of new dotted vessels that were not visible at the initial observation. This lesion was subsequently excised and diagnosed as melanoma in situ. Further changes were noted in another lesion after four months, specifically at the 12 o'clock position. In this area, marked in white, a negative network began to appear that was absent at the initial examination. This lesion was also excised and identified as an invasive melanoma, measuring only 0.5 millimeters in thickness and without any ulceration, suggesting a favorable prognosis.

**Question:** What was observed in the melanocytic lesion on the upper arm after four months of follow-up?
**Answer:** Development of new dotted vessels and fibrosis.

**Question:** What type of medical condition was diagnosed for the initial altered lesion on the upper arm?
**Answer:** Melanoma in situ.

**Question:** What visual change was noted at the 12 o'clock position in the second lesion after four months?
**Answer:** Negative network appearance.

**Question:** What was the final diagnosis of the lesion with changes at the 12 o'clock position after excision?
**Answer:** Invasive melanoma.

## Example #12

**Modality:** Ultrasound
**Anatomical Region:** Thoracic and Respiratory System
**Query:** mghultrasound8334
**Link:** https://youtu.be/A2cVp8LklRI

7:24                                                                            11:04

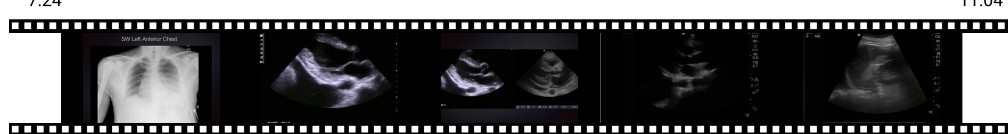

**Cleaned Caption:** This biomedical video content describes an urgent clinical scenario involving a patient experiencing hemodynamic instability due to an undetermined internal bleeding source. A portable chest X-ray has been conducted prior to the arrival of an ultrasound machine, and vital signs are deteriorating, indicating a critical situation.

In such cases, it is crucial to initially focus the ultrasound probe around the heart, particularly if the patient has sustained a stab wound to the left anterior chest, as cardiac injury can be life-threatening. The first ultrasound image displayed is a parasternal long view, revealing the left atrium, left ventricle, and aortic outflow tract, alongside the descending thoracic aorta. Adjacent to the heart, a hypoechoic fluid is observed. The main concern is whether this fluid is localized in the left chest or around the heart.

On the parasternal long view, bleeding in the right chest is not visible. The differentiation between pleural and pericardial effusions—even in non-traumatic medical patients—is determined by the fluid's location relative to the descending thoracic aorta. Fluid located posterior to this structure indicates a pleural effusion, while fluid anterior to it suggests a pericardial effusion.

The video provides a clear example of both conditions: a pleural effusion on the left side of the chest is identified posterior to the descending thoracic aorta, while a pericardial effusion is noted where fluid accumulates anteriorly. It is possible for patients to have both conditions. Again, the parasternal long view illustrates the left atrium, left ventricle, aortic outflow tract, right ventricle, and the descending thoracic aorta, with fluid observed both posteriorly and anteriorly to this vessel, indicating simultaneous pleural and pericardial effusions.

If uncertainty persists regarding fluid in the pleural cavities, further examination should include the FAST (Focused Assessment with Sonography for Trauma) view, specifically the left and right upper quadrants above the diaphragm. In this examination, no blood is detected in the abdominal Morrison's pouch, which appears normal, but an anechoic fluid is noted above the diaphragm. In a well-aerated lung, a mirror image artifact appears when ultrasound beams cross the diaphragm, creating an apparent reflection resembling the liver. This artifact disappears in the presence of fluid, consolidation, or blood, making the anechoic fluid here significant.

Additionally, a positive spine sign is observed: this white ridge along the screen's bottom is the spine. Normally, the spine is visible along...

**Question:** What does the video reveal about the location of the fluid in relation to the descending thoracic aorta?
**Answer:** Fluid is both anterior and posterior to the descending thoracic aorta, indicating pleural and pericardial effusions.

**Example #13**

**Modality:** Ultrasound
**Anatomical Region:** Cranial and Nervous System
**Query:** sonography
**Link:** https://youtu.be/RskrEsAGzec

2:40                                                                5:20

**Cleaned Caption:** The video explains the use of rocking, rotation, and tilting to enhance imaging in a narrow acoustic window. Rocking helps extend the imaging plane. Rotation switches between short and long axis imaging. Tilting enables scanning along a structure's course through the acoustic window. In nerve imaging, optimizing the transducer's tilt is crucial for visualizing nerve fascicles, which are best seen when the plane is orthogonal to the nerve's course. The focus is on short-axis nerve imaging and needle approaches.
The out-of-plane needle approach involves the needle starting in front of and crossing the imaging plane as an echogenic dot. In the in-plane approach, the needle advances along its long axis within the imaging plane. Challenges in visualizing block needles sonographically are discussed. Block needles are specular reflectors, reflecting sound back over a narrow angle. As the angle between the transducer and needle increases, less sound is reflected, weakening the signal, especially from the needle shaft's sides.
With a short-axis out-of-plane approach, the needle is best visualized when the tip is in the imaging plane, as the shaft's signal may not be stronger than surrounding structures. Generally, tissue displacement indicates the needle's trajectory, a dynamic process reliant on the tissue plane's transmission.

**Question:** What technique is demonstrated in the video for improving imaging in a narrow acoustic window?
**Answer:** Rocking, rotation, and tilting of the transducer.

**Question:** How is the needle visualized in an in-plane approach as shown in the video?
**Answer:** Along its long axis within the imaging plane.

**Question:** In the context of the video, why is optimizing the transducer's tilt crucial for nerve imaging?
**Answer:** To visualize nerve fascicles orthogonally.

**Question:** What challenge is discussed regarding the sonographic visualization of block needles?
**Answer:** They reflect sound back over a narrow angle.

## Example #14

**Modality:** MRI
**Anatomical Region:** Endocrine System
**Query:** dynamic_mri
**Link:** https://youtu.be/DsJ7Kia1eVg

2:20                                                                                  5:10

**Cleaned Caption:** The video describes the planning process for T1 sagittal magnetic resonance imaging (MRI). The sequence time slice is an odd number, and the scan time is set to be 3 minutes and 24 seconds, with no gap. First, the video demonstrates the planning of the axial view, which is essential for accurate planning of the sagittal view to ensure comprehensive coverage. The planning for T1 sagittal is then aligned with the T2 planning, maintaining a zero gap. The scan time for T2 sagittal is noted to be 2 minutes. The video also references images, including a flare axial view, and highlights the swollen pituitary gland. It mentions the upcoming dynamic pituitary contrast imaging to rule out any lesions, as well as the use of dynamic PT2D contrast. Diffusion imaging is also presented. The video encourages viewers to ask questions in the comment section if they have any queries.

**Question:** What is the scan time set for the T1 sagittal MRI as shown in the video?
**Answer:** 3 minutes and 24 seconds.

**Question:** What anatomical feature is highlighted in the video as being swollen?
**Answer:** Pituitary gland.

**Question:** How does the video describe the alignment between T1 sagittal and T2 imaging?
**Answer:** Maintaining a zero gap.

**Question:** What imaging technique is mentioned for ruling out lesions in the pituitary gland?
**Answer:** Dynamic PT2D contrast.

> ⚠ **Warning**
>
> This example contains surgical material and may not be suitable for all viewers.

## Example #15

**Modality:** Surgical and Procedural
**Anatomical Region:** Musculoskeletal System
**Query:** surgical_procedure_video
**Link:** https://youtu.be/zfrMhSebW2g

6:46                                                                    9:00

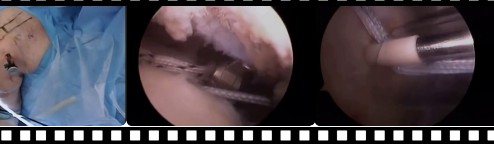

**Cleaned Caption:** The suture is pulled through a loop and subsequently into the subacromial space, allowing it to be shuttled through the rotator cuff tendon. Upon completion, all four sutures pass through the tendon, necessitating the grasping of corresponding sutures to facilitate knot tying. Two white sutures are selected. Outside the cannula, a specific sliding knot known as the SMC knot is tied. By pulling on one end of the suture, the knot is slid down into the subacromial space while the knot pusher device firmly positions it onto the rotator cuff, effectively indenting it. Half hitches, consisting of multiple knots, are tied above the initial knot to enhance security. This tying occurs outside the cannula, ensuring the knot tightens effectively, securing the rotator cuff to the anchor. Once completed, the blue sutures are retrieved, with the white sutures placed in another cannula to avoid tangling. These sutures are then pulled outside the body to tie the same knot once more. The knot pusher aids in sliding the suture down to cinch the knot firmly, emphasizing the importance of this step. Preparations for the lateral anchors begin with retrieving one blue suture and one white suture each. These are loaded outside the body into push lock anchors designed to secure the tendon over the footprint of the greater tuberosity. The anchors are seated in the robust lateral bone of the greater tuberosity. The bone is confirmed solid by our punch, and a pilot hole is made as demonstrated. The loaded push lock anchor is then placed in the pilot hole. Before seating the anchor, any slack is removed from the sutures to ensure a snug tendon position. When satisfied with the positioning, the anchor is seated, and the ends of the suture are cut.

**Question:** What type of knot is primarily used in securing the rotator cuff in the video?
**Answer:** SMC knot.

**Question:** Where are the white sutures initially positioned before tying the knot as shown in the video?
**Answer:** Outside the cannula.

**Question:** What is the purpose of the knot pusher device in the procedure shown in the video?
**Answer:** To slide the knot down and position it.

**Question:** What specific area is targeted for the placement of lateral anchors in the procedure shown in the video?
**Answer:** Greater tuberosity.

## G.2 SurgicalQABench

> ⚠ **Warning**
>
> This example contains surgical material and may not be suitable for all viewers.

---

**Example #1**

**Modality:** Surgical and Procedural
**Anatomical Region:** Cranial and Nervous System
**Query:** surgical_procedure_videos
**Link:** https://youtu.be/1gKMtSA6VCY

**Cleaned Caption:** The video gradually introduces various anatomical features, beginning with a part of the eye connected to a nerve. It then highlights the pectorals and their associated nerves, illustrating the dissection process toward a specific direction. The sarcoid dorsal pedicle is identified, along with the lateral and medial pectoral regions, demonstrating the healing approach. The long thoracic nerve is mentioned, though it has not been fully exposed due to the intent not to remove the fascia or denude the nerve or vein. The dissection technique is emphasized as crucial for maintaining the integrity of the tissues involved. The pectoralis major and latissimus dorsi muscles, along with the latissimus dorsi pedicle, have been exposed, allowing upward movement of the nodal tissue. The video concludes with a focus on the exposure of the vein and a specific vessel referred to as "Adam's Galaxy."

**Question:** What muscles are exposed during the procedure?
**Answer:** Pectoralis major and latissimus dorsi.

**Question:** What structure is identified after the pectorals?
**Answer:** Sarcoid dorsal pedicle.

> ⚠ **Warning**
>
> This example contains surgical material and may not be suitable for all viewers.

### Example #2

**Modality:** Surgical and Procedural
**Anatomical Region:** Gastrointestinal System
**Query:** surgical_procedure_videos
**Link:** https://youtu.be/TZDe6rhoV2Y

0:24                                                  4:04

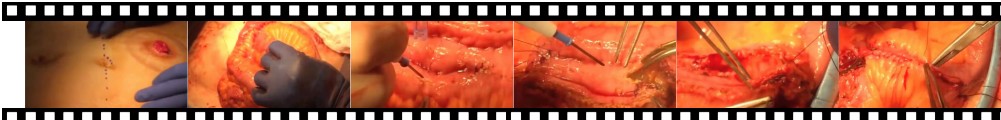

**Cleaned Caption:** The abdomen is approached via a midline laparotomy incision, revealing the previous side-to-side stapled anastomosis from the closure of the ileostomy. The mesenteric vessels are sealed and divided using a ligature, and the terminal ileum is stapled and divided just proximal to the ileocecal valve. Accurate measurement of the pouch is crucial; in this case, 15 centimeters are measured from the transected terminal ileum to ensure sufficient small bowel length for forming the nipple valve and corresponding ileostomy. An additional 15 centimeters are measured for each of the three adjacent limbs of the S pouch. A suture is inserted to mark the top of the pouch after the first 15 centimeters, with subsequent sutures marking each subsequent limb. Any discrepancies in length must be identified and corrected to ensure the limbs are of equal length.

Once all marking sutures are appropriately inserted, the first and second limbs, as well as the second and third limbs of the small bowel, are positioned adjacent and joined with a continuous 3-0 Vicryl seromuscular suture. The procedure continues with the joining of the second two adjacent limbs. Following this, diathermy is used to mark the incision line along each limb on either side of the suture line, employing the full length of the joined loops of small bowel to maximize pouch capacity. After the incision lines are marked, the small bowel is carefully incised along these lines and laid open. Each limb of the pouch is opened in succession, with care taken to avoid damaging the posterior wall of the lumen, especially when navigating corners to maintain suture integrity.

Once each loop of ileum has been laid open, the adjacent posterior walls are sutured together using a continuous 3-0 Vicryl suture, with the second and third limbs also being sutured. Upon completing the posterior wall, the anterior pouch wall is formed by suturing the lateral edges of the original first and third limbs of the S configuration. Stay sutures are inserted at each end of the pouch and in the middle of the anterior wall. Initially, only the lower half of the pouch is joined with a continuous suture, leaving the top half open to allow access for the nipple valve formation. Mesenteric fat bordering the terminal ileum is thinned out using diathermy to facilitate interception for nipple valve construction. A diathermy scratch pad is applied to rub...

**Question:** What incision is used to approach the abdomen?
**Answer:** Midline laparotomy.

**Question:** How is the terminal ileum stapled and divided?
**Answer:** Just proximal to the ileocecal valve.

**Question:** What technique is used to mark the incision lines along each limb of the small bowel?
**Answer:** Diathermy.

**Question:** How are the anterior pouch walls formed?
**Answer:** By suturing the lateral edges of the first and third limbs.

---

### ⚠ Warning

This example contains surgical material and may not be suitable for all viewers.

**Example #3**

**Modality:** Surgical and Procedural
**Anatomical Region:** Skin and Integumentary System
**Query:** surgical_procedure_videos
**Link:** https://youtu.be/lBmRC9LhFgw

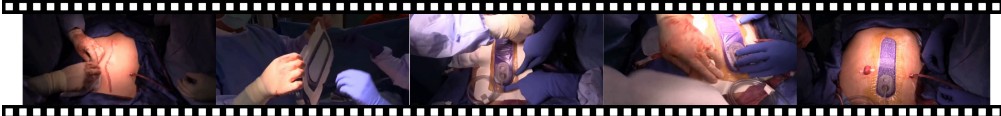

**Cleaned Caption:** To minimize contamination, benzoin is applied around the wound's border to enhance the adherence of the Pravena system. The system is then introduced into the surgical field and should be assessed to ensure it fully covers the wound area. This assessment is performed by running the blunt end of an instrument around the sponge. Next, the backing on the side sections of the dressing is removed. If a slit is necessary for the drain, it should be cut out at this point. A stoma rod is then inserted through the mesenteric defect and should be positioned anteriorly to the dressing. Suction is then applied to the completed dressing setup. The stoma is matured, and the applicable appliance is placed over the Pravena dressing. It is recommended that the system remain in place until the seventh postoperative day, after which the incision is left open to air. If the patient is ready for discharge before postoperative day seven, they may be discharged with a small vacuum canister to maintain the dressing's function.

**Question:** What is assessed to ensure full coverage of the wound area?
**Answer:** Pravena system.

**Question:** What method is used to assess the coverage of the Pravena system?
**Answer:** Running the blunt end of an instrument around the sponge.

**Question:** What needs to be inserted through the mesenteric defect?
**Answer:** Stoma rod.

## G.3 MIMICEchoBench

**Example #1**

**Example:** mimic-iv-echo/0.1/files/p16/p16673511/s90022296/90022296_0053.dcm

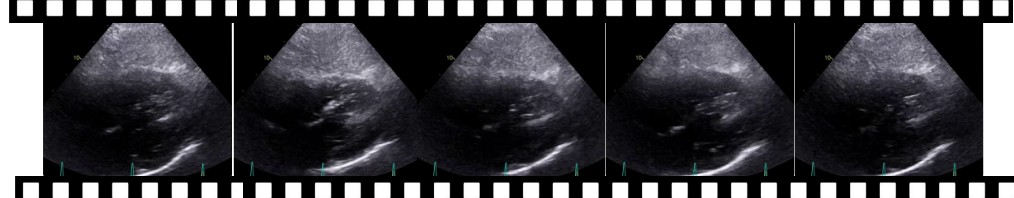

**Question:** Is there normal biventricular systolic function?
**Answer:** Yes

**Example #2**

**Example:** mimic-iv-echo/0.1/files/p12/p12246674/s99997314/99997314_0001.dcm

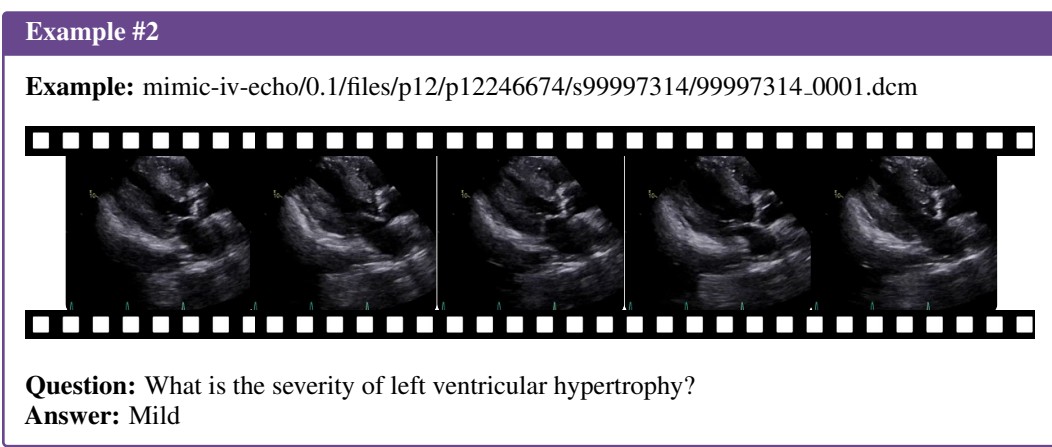

**Question:** What is the severity of left ventricular hypertrophy?
**Answer:** Mild

## G.4    SigLIP Image Classification Dataset

⚠ Warning

This example contains surgical material and may not be suitable for all viewers.

**Biomedical Images**

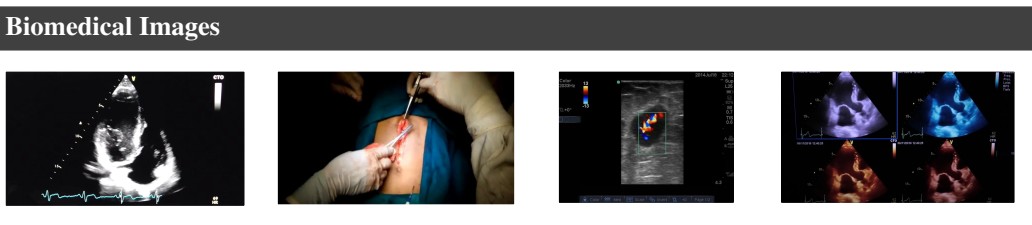

**Non-Biomedical Images**

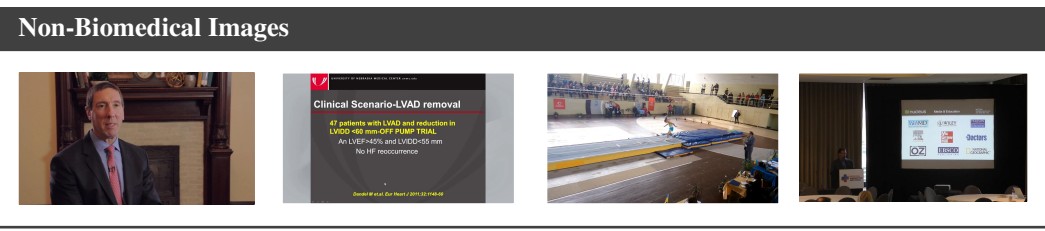

