# OpenReview forum: "How Well Can General Vision-Language Models Learn Medicine By Watching Public Educational Videos?"
_ICLR.cc/2026/Conference — Submitted to ICLR 2026_

### Official Review · Reviewer_QWpu · 2025-10-28

**Soundness:** 4
**Presentation:** 4
**Contribution:** 4
**Rating:** 6
**Confidence:** 5

**Summary:**

This paper introduces OpenBiomedVid, a biomedical video-text dataset (1,031 hours) curated from YouTube educational videos, along with two expert-curated benchmarks—SurgeryVideoQA and MIMICEchoQA—for evaluating biomedical video understanding. The authors further fine-tune Qwen2-VL and InternVL3 models on this dataset, demonstrating improvements on both video and image benchmarks.

**Strengths:**

1. Clear presentation: The paper clearly describes the construction of instruction data, evaluation datasets, and model training procedures, making the methodology easy to follow.

2. Thoughtful data curation: The dataset creation process includes several reasonable and interesting design choices, such as leveraging SigLIP-Medical and Whisper models, to ensure the reliability and quality of the collected data.

3. Practical and novel contribution: While prior works have explored YouTube data for research, the focus on video instruction tuning and the introduction of corresponding evaluation benchmarks fills an important gap in the current biomedical multimodal landscape. The dataset and benchmarks are likely to be of practical use to the community.

**Weaknesses:**

1. Although the paper targets video-level instruction tuning, prior works (e.g., Quilt-1M) have already shown success with image-level instruction data in the medical domain. It remains unclear whether video-level supervision provides a substantial advantage over image-text data for medical VQA tasks. A valuable follow-up experiment could involve creating a subset of the dataset where temporal information is minimized (e.g., few-frame clips or frame-level QA) to empirically assess whether videos or images contribute to the performance gains.
2. In lines 343–346, the authors state that “performance on video benchmarks remains significantly lower than on text and image benchmarks.” However, the reported results show only modest improvements on text benchmarks (with MedQA even decreasing) and limited gains for PathVQA with the 7B model. The discussion should be more nuanced.
3. The dataset mainly focuses on videos, but I noticed certain performance improvements on the image-text datasets VQA-RAD and SLAKE. If possible, I encourage the authors to derive and release a medical image instruction tuning dataset from the existing collection. For example, by selecting clips with fewer frames or converting segments that do not require strict temporal encoding into multi-image samples, since many QA pairs may not rely on temporal information. This would further advance the development of medical vision-language models.

**Questions:**

See weaknesses. I may consider raising my score depending on the authors’ response, as I am genuinely interested in this work.

---

> ### Author Response · Authors · 2025-11-20
> **Response Part 1**
>
> > Although the paper targets video-level instruction tuning, prior works...A valuable follow-up experiment could involve creating a subset of the dataset where temporal information is minimized (e.g., few-frame clips or frame-level QA) to empirically assess whether videos or images contribute to the performance gains.
>
> We thank the reviewer for this thoughtful suggestion and conducted the proposed experiment. As shown in Table S3, we created a “biomed-frame” variant of our dataset by selecting the highest-confidence biomedical frames (via our SigLIP classifier) and pairing them with the same instruction-tuning Q/A. This provides a direct comparison between video-level and frame-level supervision on the same backbones. We find that frame-only tuning captures a substantial portion of the signal, however, full-video supervision remains superior. For instance, on Qwen2-VL-7B, MIMICEchoQA achieves 49.0% with video-level tuning vs. 44.0% with frame-level, and SurgeryVideoQA achieves 25.1% vs. 23.0%. Following the reviewer’s recommendation—and motivated by the strong frame-only results—we will also release the image instruction-tuning subset derived from our frame-based selection.
>
>
> > In lines 343–346, the authors state that “performance on video benchmarks remains significantly lower than on text and image benchmarks.” However, the reported results show only modest improvements on text benchmarks (with MedQA even decreasing) and limited gains for PathVQA with the 7B model. The discussion should be more nuanced.
>
> We appreciate this observation and agree that our discussion should more clearly distinguish absolute accuracy from relative improvements across modalities. Our statement that “performance on video benchmarks remains significantly lower” refers to absolute accuracy: both MIMICEchoQA and especially SurgeryVideoQA remain challenging for all models, including strong baselines such as GPT-4o, Gemini, and o4-mini. In contrast, the relative gains from instruction tuning are substantially larger on video tasks (e.g., Qwen2-VL-2B: 21.1 → 42.0% and 10.3 → 20.4%), whereas improvements on text tasks are more modest and MedQA even decreases. For image benchmarks, the 7B model shows limited gains on PathVQA but meaningful improvements on VQA-RAD and SLAKE. These patterns indicate that different modalities benefit differently from the supervision signal: video tasks see large relative improvements because they begin from a low baseline, while text tasks—already well covered by pretraining—benefit less and may even experience mild misalignment due to the predominantly video-centric training corpus. We will revise the discussion to present this nuance more precisely.
>
> > The dataset mainly focuses on videos, but I noticed certain performance improvements on the image-text datasets VQA-RAD and SLAKE. If possible, I encourage the authors to derive and release a medical image instruction tuning dataset from the existing collection...
>
> We thank the reviewer for this excellent suggestion. Motivated by this feedback, we derived a medical image instruction-tuning subset from our dataset by selecting the highest-confidence biomedical frames (using our SigLIP classifier) and pairing them with the same instruction-tuning Q/A. We will release this dataset together with our video dataset.

---

> > ### Comment · Reviewer_QWpu · 2025-11-27
> > **Response**
> >
> > I have read the author's response. Considering the concerns of other reviewers, I have decided to maintain the original score.

---

> > > ### Author Response · Authors · 2025-11-28
> > >
> > > Thank you for the follow-up. Could you kindly clarify which specific concerns you feel remain unaddressed? We would be happy to address them directly and incorporate any needed improvements into the paper.

---

### Official Review · Reviewer_fKBP · 2025-10-30

**Soundness:** 2
**Presentation:** 2
**Contribution:** 2
**Rating:** 2
**Confidence:** 3

**Summary:**

The paper investigates the potential of general-purpose vision-language models to be directed towards effective biomedical video understanding by training them on publicly available educational content, primarily sourced from YouTube. The authors have constructed a substantial instruction-tuning corpus, named OpenBiomedVid, which comprises approximately 1,031 hours of clinician-guided biomedical videos, meticulously cleaned captions, and GPT-assisted yet human-verified question-answer pairs. Additionally, they introduce two more challenging expert-curated benchmarks—MIMICEchoQA (echo) and SurgeryVideoQA (surgery)—to ensure that evaluations are not conducted on the same noisy distribution as the training data. Fine-tuning Qwen2-VL (2B/7B) using this dataset results in significant relative improvements in biomedical video question answering (QA) and notable advancements in image visual question answering (VQA). In some instances, performance approaches or even surpasses that of stronger general models when applied to echo-style videos. This finding suggests that "videos made for humans" can still serve as an effective supervisory signal for medical vision-language models. However, it is important to note that results on the surgical benchmark remain distinctly lower. This indicates that long and heterogeneous procedural videos continue to pose challenges and highlights concerns regarding stylistic inconsistencies and potential hallucination risks introduced by LLM-in-the-loop curation—a concern the authors attempt to address through human verification. In summary, this work presents a well-motivated dataset and benchmark package along with compelling empirical evidence supporting the notion that public educational videos represent a viable resource for domain adaptation of open vision-language models.

**Strengths:**

- Clear Motivation & Well-Defined Problem: The authors identify an important gap: most biomedical video datasets are either small, narrowly focused, or unsuitable for large-scale multimodal learning. They correctly observe that public educational videos are a massive, untapped source for such efforts.
- Novel Dataset and Benchmark Creation: The curation of a 1031-hour biomedical video-text dataset from public sources is commendable. The pipeline is systematic, leveraging both expert clinical input and multi-stage LLM filtering. The inclusion of 79,367 Q/A pairs and structured metadata adds substantial utility and diversity.
- Expert-Curated Evaluation: The introduction of the MIMICEchoQA and SurgeryVideoQA benchmarks is a valuable contribution, offering much-needed standardized, clinically relevant tests for biomedical VLMs. The expert review and curation of questions are a particular highlight.

**Weaknesses:**

The paper creates a same-model, same-style loop: GPT-4o is used to clean captions and generate the training/eval Q&A and metadata, and then the very same GPT-4o is used as the automatic judge to score open-ended answers (binary 0/1). The authors even note the risk of “stylistic bias” and only add a one-off check with Gemini-2.0-Flash, but GPT-4o remains the primary scorer. This tightly couples data generation and evaluation to one model’s style, making gains plausibly evaluation-protocol-driven rather than true capability.

- Lack of evaluation against domain-specific medical VLMs. The paper compares mainly with general-purpose multimodal backbones (Qwen2-VL at different scales, InternVL3-8B, GPT-4o, Gemini-2.0-Flash, etc.) on the proposed biomedical video QA benchmarks, but does not report results for the most directly relevant medical instruction-tuned VLMs such as LLaVA-Med, Med-Flamingo (medical adaptation of Flamingo), or widely used medical multi-image models (e.g., MedCLIP), even though many of them are cited in Related Work. Because these models target clinical/biomedical vision-language understanding on VQA-RAD, PathVQA, SLAKE and related tasks, including at least a frame-based or image-only adaptation of them on the authors’ benchmarks would make the claimed gains over “prior medical VLMs” easier to judge. The current tables therefore make it hard to tell whether the improvement comes from the proposed video-centric data and pipeline or simply from using stronger general backbones. (We acknowledge that some models, e.g. MedCLIP or MedGemini, are image-centric or not fully open, but even a best-effort comparison or a discussion of feasibility would strengthen the evaluation.)
- Reliance on LLM-as-a-judge without human grounding. For the open-ended SurgeryVideoQA benchmark, the paper evaluates all models with GPT-4o as the automatic grader, assigning binary correctness against the reference answer. Since GPT-4o was also involved in earlier stages of caption refinement and QA generation, this creates a potential style / phrasing bias toward GPT-4o-like answers. The authors do run a second pass with Gemini-2.0-Flash and report that the ranking is broadly consistent, which is helpful, but both judges are frontier LLMs with unknown overlap in training data and alignment objectives, and no human adjudication, inter-rater agreement, or adversarial stress tests are provided. As a result, part of the reported gains on SurgeryVideoQA could still be influenced by judge-specific preferences rather than purely by better video understanding.
- Potential residual data-quality and provenance issues. Although the paper presents a multi-stage, human-in-the-loop curation pipeline (YouTube retrieval → GPT-labeled frame filtering with a fine-tuned SigLIP → GPT-4o caption refinement → GPT-4o Q/A generation → human verification) and even reports ~95% agreement on the frame-filtering stage, many of the quality controls are described only at a high level. In particular, the paper does not spell out how deep the human verification went (per-clip vs per-QA, random spot checks vs full passes), how many annotators were involved, or what the inter-annotator agreement was beyond the small sample quoted. Moreover, because GPT-4o is used both to clean captions and to synthesize Q/A pairs, typical LLM failure modes—overly generic answers, temporal misalignment with the actual video segment, or medical hallucination—are plausible but not systematically audited or reported. This matters especially because the paper itself shows that models trained on the noisy YouTube-derived corpus still perform noticeably worse on the cleaner, expert-curated benchmarks (MIMICEchoQA, SurgeryVideoQA), suggesting that remaining noise in the large-scale training split may be a limiting factor. A more explicit error analysis (e.g., failed frame segmentation, ambiguous captions, low-quality or hallucinated Q/A) would make the dataset contribution stronger.
- Underdeveloped video baselines on the proposed benchmarks. Because the core claim of the paper is about video-centric biomedical understanding, the experimental setup on MIMICEchoQA and SurgeryVideoQA would be more convincing if it included stronger, publicly available video–language models beyond the authors’own Qwen2-VL fine-tunes. At minimum, prior general-purpose video chat models (e.g., Video-ChatGPT, Video-LLaVA, PALIGemma-style video variants) could be run in a frame-sampling or short-clip regime to provide external points of reference, even if they are not domain-tuned. Their absence makes the reported gains look partly relative to the authors’ chosen baselines rather than to the broader video–language landscape, and it also hides whether the new benchmarks are genuinely “hard” for off-the-shelf video models or only for image-first VLMs.

**Questions:**

1. You cite several domain-specific multimodal/medical VLMs in Related Work (e.g., LLaVA-Med, Med-Flamingo, MedCLIP), but they do not appear in the main comparison tables. Can you clarify which of these models you actually attempted to run, and what prevented you from including their results?

2. Right now the improvements are mostly against Qwen2-VL/InternVL-style backbones. How can we be sure the gains come from your video-centric data/pipeline rather than simply from using a stronger base model?

3. The evaluation of SurgeryVideoQA was conducted using GPT-4o as the automatic grading system. Given that GPT-4o was also employed for the refinement of captions and question-and-answer pairs, what measures were taken to mitigate potential style bias towards outputs resembling those generated by GPT-4o?

4. You mention using Gemini-2.0-Flash as a second judge and finding similar rankings. Could you provide the agreement numbers between GPT-4o and Gemini on this benchmark?

5. You report ~95% agreement on frame filtering, but on how many samples, with how many annotators, and at which stage of the pipeline was this measured (video-level vs clip-level vs QA-level)?

6. Is human verification applied to every GPT-generated Q/A pair, or only to a sampled subset? If sampled, what was the sampling strategy and coverage?

7. Would it be feasible to run them in a constrained setting (e.g., fixed frame sampling, short 8–16 frame clips) just to position your benchmarks relative to the broader video-VLM ecosystem?

---

> ### Author Response · Authors · 2025-11-20
> **Response Part 1**
>
> Response to Weaknesses
>
> > The paper creates a same-model, same-style loop...
>
> We appreciate the concern regarding potential stylistic or evaluator bias introduced by GPT-4o. To rigorously assess this, we conducted a comprehensive multi-judge analysis using both GPT-4o and Gemini-2.0-Flash across all 14 evaluated models. As shown in Appendix Table S1, accuracy estimates under the two judges are highly consistent, with overlapping 95% CIs, 93.0% raw agreement, and Cohen’s kappa = 0.748. Model rankings remain nearly identical (Spearman correlation = 0.972, Pearson r = 0.993).
> To ensure evaluator independence, we also carried out a human study on 400 randomly sampled items across GPT-4o, Gemini-2.0-Flash, and our 2B/7B biomedical Qwen2-VL models. Both LLM judges show strong alignment with humans (90–95%, Cohen’s kappa = 0.72–0.84 for GPT-4o; 89–95%, Cohen’s kappa = 0.66–0.83 for Gemini), and also agree closely with each other (88–95%, Cohen’s kappa = 0.65–0.85).
>
> Finally, we highlight that GPT-4o is not generating captions de novo—the captions originate from the original YouTube audio; the LLM only cleans them and formulates Q/A pairs. Every stage of the pipeline includes human verification to prevent artifacts, and the entire SurgeryVideoQA and MIMICEchoQA benchmarks were fully reviewed by clinical experts. Using LLMs for caption refinement and QA generation is also standard practice in prior work, including LLaVA-Med (NeurIPS 2023) and Video-ChatGPT (ACL 2024), which use very similar pipelines.
>
> > Lack of evaluation against domain-specific medical VLMs...
>
> We thank the reviewer for this suggestion. Like reviewer suggested, we tried using biomedical VLM such as LLaVA-Med, however, the cited medical VLMs (LLaVA-Med, Med-Flamingo, MedCLIP) are single-image models and do not accept multi-frame video inputs or temporal sequences. Our benchmarks require reasoning over hundreds to thousands of frames, making it infeasible to adapt these image-only architectures without major architectural changes (e.g., adding video encoders, frame fusion mechanisms, or temporal attention).
>
> To directly address whether improvements come from stronger base models or from our dataset, we report detailed before/after instruction-tuning results for each Qwen2-VL backbone. Please see Figure 4. The gains are substantial and consistent across scales (e.g., +98.7% and +40.5% for 2B and 7B on video tasks), demonstrating that the improvement is derived from OpenBiomedVid, not merely from model capacity. We also compare against a broad set of general-domain multimodal models (e.g., InternVL, Phi-3.5/4, Video-ChatGPT, Video-LLaVA; Table 1), providing additional context for where our fine-tuned models stand in the current landscape.
> Finally, we clarify that this is fundamentally a dataset and benchmark paper. Our primary goal is to introduce the first large-scale biomedical video instruction-tuning dataset and two expert-curated video QA benchmarks. Within this scope, we evaluate models capable of processing video, and show improvement on expert curated biomedical video benchmarks.
>
> > Reliance on LLM-as-a-judge without human grounding...
>
> Thank you for raising this concern. We kindly refer the reviewer to our detailed response to Limitation #1, where we specifically address evaluator bias. Briefly, our conclusions on SurgeryVideoQA are not judge-dependent: GPT-4o and Gemini-2.0-Flash produce highly consistent accuracy estimates (overlapping 95% CIs, 93% raw agreement, κ = 0.748) and nearly identical model rankings (ρ = 0.972). Most importantly, we conducted a human adjudication study on 400 sampled examples across all model families. Both LLM judges show strong alignment with human reviewers (90–95% agreement; κ = 0.72–0.84 for GPT-4o and 0.66–0.83 for Gemini), and also agree closely with each other (κ = 0.65–0.85).
>
> > Underdeveloped video baselines on the proposed benchmarks...
>
> We appreciate the reviewer’s suggestion and have expanded our baseline comparisons in Table 1 to include two widely used open-source video-language models—Video-ChatGPT and Video-LLaVA—in addition to the models previously reported. Table 1 also already included several recent and competitive open-source multimodal models capable of video understanding, such as InternVideo2.5-Chat-8B, Phi-3.5-vision-instruct, and Phi-4-multimodal-instruct. Across both benchmarks, the off-the-shelf video models achieve modest accuracy (e.g., 7.9–16.4% on SurgeryVideoQA), confirming that the tasks are genuinely challenging for existing video-language systems rather than only for our chosen backbones. Fine-tuning Qwen2-VL on OpenBiomedVid yields substantial improvements over all video baselines (e.g., Qwen2-VL-2B-biomed: 42.0% / 20.4%; Qwen2-VL-7B-biomed: 49.0% / 25.1%), demonstrating that the performance gains come from the video-centric educational dataset, not merely from model strength.

---

> > ### Author Response · Authors · 2025-11-20
> > **Response Part 2**
> >
> > > Potential residual data-quality and provenance issues...
> >
> > We thank the reviewer for highlighting the importance of data-quality transparency. We clarify that human verification occurred at different depths for the training dataset versus the evaluation benchmarks. For the training dataset, human review was performed through random spot checks: a medical doctor inspected ~1,000 sampled clips together with all associated Q/A pairs, covering a broad range of channels and content areas. This sampling strategy was intentional—the goal of the paper is to test whether models can learn from naturally noisy and heterogeneous educational videos, so we did not exhaustively clean the training corpus. In contrast, human verification for the evaluation benchmarks was conducted at the per-clip and per-QA level (i.e., full passes). MIMICEchoQA was fully curated by a board-certified cardiologist, and SurgeryVideoQA was fully curated by a general medical doctor; both experts removed any items showing common LLM failure modes such as generic answers, incorrect grounding, temporal mismatch, or hallucinated clinical content.
> >
> > Regarding annotator counts, each benchmark involved two clinical annotators, and the frame-filtering stage involved two independent annotators, where we observed 95% human–model agreement and 93% human–human agreement on 1,000 frame-level samples immediately after training the SigLIP classifier. We agree that LLM-generated training captions/QAs may still contain occasional artifacts—this is inherent to the educational YouTube domain and aligned with the research question of whether models can learn effectively from such heterogeneous supervision.
> >
> > To better quantify residual noise in the training corpus, we conducted an manual analysis of 100 randomly sampled caption–QA pairs, labeling each item with its primary issue when present. We found that 15% of samples contained ambiguous or low-information captions (e.g., generic openings such as “In this video…” that provide little clinical grounding), and 10% showed signs of incorrect grounding, where the caption was vague but the QA referred to specific anatomy or procedures. Only 5% exhibited clear LLM-style noisy phrasing (e.g., “the example under consideration…”). Importantly, 70% of samples were mostly normal and clinically grounded, consistent with our design choice to preserve some natural heterogeneity in the training data. As emphasized earlier, the two evaluation benchmarks—MIMICEchoQA and SurgeryVideoQA—are fully expert-curated and do not contain these artifacts. We will include representative cases in an error-analysis appendix to further document these patterns.
> >
> > Response to Questions:
> >
> > > You cite several domain-specific multimodal/medical VLMs in Related Work (e.g., LLaVA-Med, Med-Flamingo, MedCLIP), but they do not appear in the main comparison tables. Can you clarify which of these models you actually attempted to run, and what prevented you from including their results?
> >
> > We have answered the question above in response to the Limitation section. Briefly, these are single image VLM models, whereas our benchmarks are video with multiple image frames.
> >
> > > Right now the improvements are mostly against Qwen2-VL/InternVL-style backbones. How can we be sure the gains come from your video-centric data/pipeline rather than simply from using a stronger base model?
> >
> > We agree that isolating the source of improvements is important. To make this explicit, all of our core results compare the same backbone before and after fine-tuning on OpenBiomedVid. The gains are therefore attributable to the video instruction-tuning data, not to model capacity. For example, Qwen2-VL-2B improves from 21.1% → 42.0% on MIMICEchoQA and 10.3% → 20.4% on SurgeryVideoQA after training on our dataset; similarly, the 7B model improves from 37.9% → 49.0% and 16.5% → 25.1%, respectively. These substantial deltas on the same architecture demonstrate that the improvements arise from the educational video data, not from using a stronger backbone. Moreover, Table 1 evaluates a broad set of publicly available video-capable VLMs—including Video-ChatGPT, Video-LLaVA, InternVideo2.5-Chat-8B, Phi-3.5-vision, Phi-4-multimodal, GPT-4o, Gemini-2.0-Flash, and o4-mini—and the fine-tuned models outperform most of them on average.
> >
> > > The evaluation of SurgeryVideoQA was conducted using GPT-4o as the automatic grading system. Given that GPT-4o was also employed for the refinement of captions and question-and-answer pairs, what measures were taken to mitigate potential style bias towards outputs resembling those generated by GPT-4o?
> >
> > We respond to this query in our response to your limitation section.
> >
> > > You mention using Gemini-2.0-Flash as a second judge and finding similar rankings. Could you provide the agreement numbers between GPT-4o and Gemini on this benchmark?
> >
> > Yes, we provide the agreement between GPT-4o and Gemini above as well. Please check our response to your limitation section above.

---

> > > ### Author Response · Authors · 2025-11-20
> > > **Response Part 3**
> > >
> > > > You report ~95% agreement on frame filtering, but on how many samples, with how many annotators, and at which stage of the pipeline was this measured (video-level vs clip-level vs QA-level)?
> > >
> > > The ~95% agreement figure refers to our frame-level validation of the SigLIP biomedical–non-biomedical classifier. We sampled 1,000 frames, and two independent human annotators labeled each as biomedical or not. Both annotators showed ≈95% agreement with the SigLIP classifier, and their inter-annotator agreement was ≈93%. This validation was performed immediately after training the classifier, as depicted in Figure 1, to ensure that the frame-selection stage of the pipeline was reliable before generating captions or Q/A pairs. Beyond this initial filtering step, we also incorporated human review at multiple later stages: a medical doctor spot-checked 1000+ randomly sampled training videos (video- and QA-level), and both evaluation benchmarks were fully curated by clinical experts—a general medical doctor for SurgeryVideoQA and a board-certified cardiologist for MIMICEchoQA—who removed noisy or unanswerable items. The training dataset intentionally preserves some natural heterogeneity, but the evaluation datasets are entirely expert-cleaned, ensuring high-quality test sets.
> > >
> > > > Is human verification applied to every GPT-generated Q/A pair, or only to a sampled subset? If sampled, what was the sampling strategy and coverage?
> > >
> > > For the training dataset, human verification was applied to a sampled subset, not every GPT-generated Q/A pair, due to the scale of the corpus. A medical doctor randomly spot-checked more than 1000 videos (including their captions and Q/A pairs), sampling uniformly across topics. This review focused on identifying systematic issues—e.g., unanswerable questions, incorrect grounding, or irrelevant content—rather than exhaustively editing every item. In contrast, for the evaluation benchmarks, human verification was applied to every single Q/A pair. MIMICEchoQA was fully curated by a board-certified cardiologist, and SurgeryVideoQA was fully curated by a general medical doctor. These experts reviewed all videos and Q/A pairs, removing noisy or unanswerable items and ensuring clean gold labels.
> > >
> > > > Would it be feasible to run them in a constrained setting (e.g., fixed frame sampling, short 8–16 frame clips) just to position your benchmarks relative to the broader video-VLM ecosystem?
> > >
> > > We thank the reviewer for this thoughtful suggestion and conducted the proposed experiment. As shown in Table S3, we created a “biomed-frame” variant of our dataset by selecting the highest-confidence biomedical frames (via our SigLIP classifier) and pairing them with the same instruction-tuning Q/A. This provides a direct comparison between video-level and frame-level supervision on the same backbones. We find that frame-only tuning captures a substantial portion of the signal, however, full-video supervision remains superior. For instance, on Qwen2-VL-7B, MIMICEchoQA achieves 49.0% with video-level tuning vs. 44.0% with frame-level, and SurgeryVideoQA achieves 25.1% vs. 23.0%.

---

> > > > ### Author Response · Authors · 2025-11-28
> > > >
> > > > Dear reviewer,
> > > >
> > > > As the discussion period is nearing its end, we wanted to check whether our responses have sufficiently addressed your concerns, or if there are any remaining points that would benefit from further clarification. Thank you again for your time, thoughtful feedback, and engagement with our work.

---

> > > > > ### Comment · Reviewer_fKBP · 2025-11-28
> > > > >
> > > > > Dear Authors,
> > > > >
> > > > > Thanks for the detailed response and additional experiments. They address my concerns to some extent. I’d like to increase my score, but the system seems temporarily locked. If it reopens for score updates, please reach out and I’ll revise my score.

---

> > > > > > ### Author Response · Authors · 2025-11-28
> > > > > >
> > > > > > Dear Reviewer,
> > > > > >
> > > > > > Thank you again for your thoughtful feedback and for considering an updated score for our submission -- we really appreciate your engagement.
> > > > > >
> > > > > > From our side, the OpenReview interface currently appears to allow score updates. If you still encounter issues editing your rating, please let us know and we will be happy to check with the area chairs / program committee about the status of score updates on your behalf.
> > > > > >
> > > > > > If there are any remaining points where our responses were unclear or did not fully address your concerns, we would also be grateful for further guidance and are happy to clarify or provide additional details.

---

### Official Review · Reviewer_bp18 · 2025-10-31

**Soundness:** 3
**Presentation:** 3
**Contribution:** 3
**Rating:** 4
**Confidence:** 4

**Summary:**

This paper investigates whether general-purpose vision-language models can effectively learn biomedical knowledge from publicly available educational videos on platforms like YouTube. It introduces OpenBiomedVid, a large-scale, diverse instruction-tuning dataset comprising 1,031 hours of biomedical video content, curated through a multi-step human-in-the-loop pipeline that filters frames, refines captions, and generates question-answer pairs. For evaluation, two new benchmarks are released: Surgery VideoQA and MIMICEchoQA.

**Strengths:**

The effort to create these resources is commendable. The field lacks large-scale biomedical video datasets, and OpenBiomedVid, along with the two new benchmarks, represents a significant contribution.

**Weaknesses:**

The paper is undermined by several major flaws in its methodology and results, which make the central claim (that the models are "learning medicine") unconvincing.

1.	Potential Data Contamination and LLM Bias: A major concern is the pervasive use of GPT-4o throughout the pipeline (frame annotation, caption refinement, Q/A generation, and evaluation). Although the authors implement safeguards, there is a non-trivial risk of the model learning a "stylistic bias" or patterns specific to GPT-4o's output, rather than genuine biomedical reasoning. The evaluation with Gemini mitigates this but does not fully eliminate the concern, as the fine-tuned models' training data was itself shaped by GPT-4o.

2.	Absolute Performance is Still Low: Despite impressive relative gains, the absolute performance on the video benchmarks remains low (e.g., 25.1% for the 7B model on SurgeryVideoQA). The paper correctly notes that these tasks are challenging, but the low scores highlight that the problem is far from solved and that the models are not yet reliable. This should be more prominently discussed in the context of the claim that models can "learn medicine."

3.	The paper claims the models are "learning medicine," but their performance on the standard MedQA text benchmark significantly decreased after fine-tuning. For example, the Qwen-2-VL-7B-Biomed model’s accuracy dropped from a baseline of 52.6% to 47.4%. This suggests the informal, "noisy" knowledge from the videos may be conflicting with, or causing the model to "forget," formal textbook knowledge. This directly contradicts the paper's primary narrative.

4.	Limited Analysis of "Why" and Failure Modes: The paper excellently demonstrates that the method works but provides less insight into why and how it fails. A deeper analysis of the types of questions or video segments where the model struggles (e.g., temporal reasoning in long surgical videos vs. static frame understanding in echocardiograms) would be highly valuable. The qualitative examples are good but are primarily success cases.

5.	Clarity on Training-Test Split and Overlap: While the authors state there is no video ID overlap between OpenBiomedVid and SurgeryVideoQA, both are sourced from YouTube. There is a potential for concept or stylistic overlap. A more detailed discussion on how the "cleanliness" and focus of the evaluation set differ from the training data would clarify the generalization claim.

**Questions:**

please see the weakness

---

> ### Author Response · Authors · 2025-11-20
> **Response Part 1**
>
> We thank the reviewer for taking the time to assess our paper and provide insightful feedback. Below, we address each of the reviewer’s concerns point by point.
>
> > Potential Data Contamination and LLM Bias...
>
> We appreciate the concern regarding potential stylistic or evaluator bias introduced by GPT-4o. To rigorously assess this, we conducted a comprehensive multi-judge analysis using both GPT-4o and Gemini-2.0-Flash across all 14 evaluated models. As shown in Appendix Table S1, accuracy estimates under the two judges are highly consistent, with overlapping 95% CIs, 93.0% raw agreement, and Cohen’s kappa = 0.748. Model rankings remain nearly identical (Spearman correlation = 0.972, Pearson r = 0.993).
> To ensure evaluator independence, we also carried out a human study on 400 randomly sampled items across GPT-4o, Gemini-2.0-Flash, and our 2B/7B biomedical Qwen2-VL models. Both LLM judges show strong alignment with humans (90–95%, Cohen’s kappa = 0.72–0.84 for GPT-4o; 89–95%, Cohen’s kappa = 0.66–0.83 for Gemini), and also agree closely with each other (88–95%, Cohen’s kappa = 0.65–0.85).
>
> Finally, we highlight that GPT-4o is not generating captions de novo—the captions originate from the original YouTube audio; the LLM only cleans them and formulates Q/A pairs. Every stage of the pipeline includes human verification to prevent artifacts, and the entire SurgeryVideoQA and MIMICEchoQA benchmarks were fully reviewed by clinical experts. Using LLMs for caption refinement and QA generation is also standard practice in prior work, including LLaVA-Med (NeurIPS 2023) and Video-ChatGPT (ACL 2024), which use very similar pipelines.
>
> > Absolute Performance is Still Low... This should be more prominently discussed in the context of the claim that models can "learn medicine."
>
> We agree with the reviewer that absolute performance on SurgeryVideoQA is still low and that these models are not yet reliable for clinical use. We fully agree that the phrase “learn medicine” may be overstating our contributions, and we are happy to revise the title and text to reflect it more accurately. Our goal is not to claim that we have solved biomedical video understanding; rather, this is precisely why we frame the work as a dataset and benchmark contribution. By releasing a large-scale training corpus and two expert-curated evaluation benchmarks, we aim to establish the foundation and infrastructure needed for future progress on this challenging problem. Importantly, the low absolute scores underscore exactly why such standardized benchmarks are needed: real surgical and echocardiography videos require nuanced temporal, anatomical, and procedural reasoning that today’s VLMs struggle with. Our main empirical finding is that noisy, heterogeneous educational videos nonetheless produce substantial improvements and cross-domain generalization, not that the problem is solved.
>
> > The paper claims the models are "learning medicine," but their performance on the standard MedQA text benchmark significantly decreased after fine-tuning...
>
> We agree with the reviewer that the 7B model shows a modest decrease on MedQA (52.6% → 47.4%). One potential explanation is that fine-tuning primarily on video–text instructional pairs may shift the model’s capacity toward multimodal grounding and away from the highly structured, textbook-style reasoning required by MedQA. Because our supervision signal is dominated by visually grounded educational content rather than text-only medical QA, some degree of tradeoff may occur. Importantly, this effect is not universal across text benchmarks: on PubMedQA, both the 2B and 7B models improve (e.g., 67.7% → 73.8% for the 7B model), showing that video-based instruction tuning does not inherently degrade textual biomedical reasoning. We have revised our contributions to accurately reflect these nuances.
>
> > Clarity on Training-Test Split and Overlap...
>
> We confirm that there is no overlap in video IDs or YouTube channels between OpenBiomedVid and our two evaluation benchmarks (MIMICEchoQA and SurgeryVideoQA). The training corpus consists of heterogeneous YouTube educational content—lectures, animations, mixed-quality recordings, diagrams, talking-head explanations, and occasional live demonstrations—which is stylistically diverse and largely informal. In contrast, the evaluation sets are substantially cleaner, more focused, and fully expert-curated. MIMICEchoQA is derived from the MIMIC-IV-ECHO institutional dataset, containing short, standardized clinical echocardiography loops paired with expert-written questions, all reviewed by a board-certified cardiologist. SurgeryVideoQA consists of long-form surgical procedures requiring temporal and procedural reasoning, with all videos and Q/A pairs manually validated by a medical doctor. Unlike OpenBiomedVid, where Q/A pairs are mostly LLM-generated, both benchmarks use human-verified questions and gold answers.

---

> > ### Author Response · Authors · 2025-11-20
> > **Response Part 2**
> >
> > > Limited Analysis of "Why" and Failure Modes...
> >
> > We thank the reviewer for this insightful comment, and in response we conducted a comprehensive quantitative and qualitative error analysis of the model trained on our dataset (Qwen2-VL-7B-Biomed-Video) across both evaluation benchmarks. For MIMICEchoQA, we grouped errors by anatomical structure and ultrasound view and found that mistakes disproportionately concentrate in regions such as left atrium (68.4% error), pulmonary artery (66.7%), and right atrium (72.7%). Doppler and PSAX/PLAX great-vessel views also exhibit the highest error (e.g., DOPPLER: 83%; PSAX great-vessels: 70–71%), consistent with the need for dynamic spectral reasoning. A manual review of 100 incorrect predictions further revealed clear, non-random patterns: roughly 60% of errors arise from severity miscalibration, where mild or severe findings are collapsed to moderate; 25–30% stem from threshold ambiguities (e.g., under-calling small pericardial effusions or over-calling physiologic aortic regurgitation); and ~10% reflect systematic Ejection Fraction overestimation. These error modes likely arise from the inherent noise and subjectivity in report-derived labels (e.g., borderline severity categories, inter-expert disagreement, and variability in clinical phrasing).
> >
> > For SurgeryVideoQA, we manually examined 100 randomly sampled incorrect predictions and again observed consistent, interpretable failure categories. Approximately 22% of errors involve anatomy/localization mistakes (e.g., confusing strap muscles with pectoralis major/minor, or pre-sacral space with the left iliac fossa). Instrument/technique/suture misidentification accounts for 25–33% of failures (e.g., predicting diathermy instead of a surgical knife). Procedural step or temporal confusion explains another ~16%, where the model describes an earlier or later operative phase rather than the interval queried. Around 10% of errors arise from overly generic responses (e.g., “a knot” instead of “purse-string notch”), and 5–10% involve quantitative or extent misestimation (e.g., dissection distance, graft volume). Across all categories, most mistakes are near misses—the model generally identifies the correct procedure and high-level context but lacks the fine-grained, expert-level specificity required by the benchmark. These patterns are consistent with residual noise and label coarseness in the large, automatically curated YouTube-derived dataset.

---

> > > ### Author Response · Authors · 2025-11-28
> > >
> > > Dear reviewer,
> > >
> > > As the discussion period is nearing its end, we wanted to check whether our responses have sufficiently addressed your concerns, or if there are any remaining points that would benefit from further clarification. Thank you again for your time, thoughtful feedback, and engagement with our work.

---

### Official Review · Reviewer_QzV3 · 2025-10-31

**Soundness:** 3
**Presentation:** 2
**Contribution:** 2
**Rating:** 6
**Confidence:** 3

**Summary:**

This paper presents **OpenBiomedVid**, a large-scale dataset of **1,031 hours of biomedical educational videos** from YouTube, containing **22K clips** and **79K Q/A pairs**. The dataset is created through a **human-in-the-loop pipeline** that combines Whisper transcription, GPT-4o caption refinement, and expert verification. The authors also introduce two benchmarks, **SurgeryVideoQA** and **MIMICEchoQA**, for biomedical video-language evaluation. Fine-tuning **Qwen2-VL (2B/7B)** models on OpenBiomedVid yields strong performance gains across biomedical video and image benchmarks, showing that open educational videos can be an effective training signal.

**Strengths:**

1. Addresses an underexplored area: **biomedical video-language modeling**.
2. The dataset is **large, diverse, and well-organized**, potentially a valuable community contribution.
3. The **data curation pipeline** is systematic, combining LLM-based processing with human verification.
4.  Empirical results are consistent across multiple benchmarks and architectures.

**Weaknesses:**

1.  **Evaluation bias** — heavy dependence on GPT-4o for caption refinement, Q/A generation, and evaluation creates potential bias and reproducibility concerns.
2. **Lack of deep analysis** — minimal discussion of failure cases, temporal reasoning, or cross-domain generalization.
3. **Ethical and legal clarity** — the discussion of data licensing, PHI risk, and content ownership is insufficient.
4. **Incremental insight** — while scale is impressive, the core finding (“educational videos help”) feels somewhat intuitive and under-analyzed.

**Questions:**

1. **Will the dataset and benchmarks be fully released?** If so, under what license and with what access restrictions?
2. How is **personal or sensitive information** (e.g., PHI, identifiable subjects) detected and filtered in the YouTube videos?
3. Does the fine-tuned model show **transferability** to unseen clinical or institutional datasets?
4. What is the ratio of educational diagrams/narration-only videos to true clinical imaging?
5. Are there **quantitative measures of data quality**, such as annotation agreement or verification accuracy?

---

> ### Author Response · Authors · 2025-11-20
> **Response Part 1**
>
> We thank the reviewer for taking the time to evaluate our work and for providing valuable feedback. Below, we address each concern.
>
> > Evaluation bias — heavy dependence on GPT-4o for caption refinement, Q/A generation, and evaluation creates potential bias and reproducibility concerns.
>
> We appreciate the concern regarding potential stylistic or evaluator bias introduced by GPT-4o. To assess this, we conducted a comprehensive multi-judge analysis using both GPT-4o and Gemini-2.0-Flash across all 14 evaluated models. As shown in Supplementary Table S1, accuracy estimates under the two judges are highly consistent, with overlapping 95% CIs, 93.0% raw agreement, and Cohen’s kappa = 0.748. Model rankings remain nearly identical (Spearman correlation = 0.972, Pearson r = 0.993).
>
> To ensure evaluator independence, we also carried out a human study on 400 randomly sampled items across GPT-4o, Gemini-2.0-Flash, and our 2B/7B biomedical Qwen2-VL models (Supplementary Table S4). Both LLM judges show strong alignment with humans (90–95%, Cohen’s kappa = 0.72–0.84 for GPT-4o; 89–95%, Cohen’s kappa = 0.66–0.83 for Gemini), and also agree closely with each other (88–95%, Cohen’s kappa = 0.65–0.85).
>
> Finally, we highlight that GPT-4o is not generating captions de novo—the captions originate from the original YouTube audio; the LLM only cleans them and formulates Q/A pairs. Every stage of the pipeline includes human verification to prevent artifacts, and the entire SurgeryVideoQA and MIMICEchoQA benchmarks were fully reviewed by clinical experts. Using LLMs for caption refinement and QA generation is also standard practice in prior work, including LLaVA-Med (NeurIPS 2023) and Video-ChatGPT (ACL 2024), which use very similar pipelines.
>
> > Lack of deep analysis — minimal discussion of failure cases, temporal reasoning, or cross-domain generalization.
>
> We thank the reviewer for this insightful comment, and in response we conducted a comprehensive quantitative and qualitative error analysis of the model trained on our dataset (Qwen2-VL-7B-Biomed-Video) across both evaluation benchmarks. For MIMICEchoQA, we grouped errors by anatomical structure and ultrasound view and found that mistakes disproportionately concentrate in regions such as left atrium (68.4% error), pulmonary artery (66.7%), and right atrium (72.7%). Doppler and PSAX/PLAX great-vessel views also exhibit the highest error (e.g., DOPPLER: 83%; PSAX great-vessels: 70–71%), consistent with the need for dynamic spectral reasoning. A manual review of 100 incorrect predictions further revealed clear, non-random patterns: roughly 60% of errors arise from severity miscalibration, where mild or severe findings are collapsed to moderate; 25–30% stem from threshold ambiguities (e.g., under-calling small pericardial effusions or over-calling physiologic aortic regurgitation); and ~10% reflect systematic Ejection Fraction overestimation. These error modes likely arise from the inherent noise and subjectivity in report-derived labels (e.g., borderline severity categories, inter-expert disagreement, and variability in clinical phrasing).
>
> For SurgeryVideoQA, we manually examined 100 randomly sampled incorrect predictions and again observed consistent, interpretable failure categories. Approximately 22% of errors involve anatomy/localization mistakes (e.g., confusing strap muscles with pectoralis major/minor, or pre-sacral space with the left iliac fossa). Instrument/technique/suture misidentification accounts for 25–33% of failures (e.g., predicting diathermy instead of a surgical knife). Procedural step or temporal confusion explains another ~16%, where the model describes an earlier or later operative phase rather than the interval queried. Around 10% of errors arise from overly generic responses (e.g., “a knot” instead of “purse-string notch”), and 5–10% involve quantitative or extent misestimation (e.g., dissection distance, graft volume). Across all categories, most mistakes are near misses—the model generally identifies the correct procedure and high-level context but lacks the fine-grained, expert-level specificity required by the benchmark. These patterns are consistent with residual noise and label coarseness in the large, automatically curated YouTube-derived dataset.

---

> ### Author Response · Authors · 2025-11-20
> **Response Part 2**
>
> > Ethical and legal clarity — the discussion of data licensing, PHI risk, and content ownership is insufficient.
>
> We thank the reviewer for raising these important points. Extended discussion of ethics is included in Appendix Section A.  Briefly, our dataset only contains YouTube URLs and derived text annotations (filtered captions and Q/A pairs); we do not redistribute any video content, thumbnails, or frames. All videos were publicly accessible educational sources, and our usage follows the same practice adopted in recent NeurIPS datasets such as MiraData (NeurIPS D&B 2024) and Mr. HiSum (NeurIPS D&B 2023), which also use publicly available YouTube videos while releasing only URLs and structured annotations. To mitigate PHI risk, we (1) performed manual review across multiple stages of the pipeline, and (2) ensured that both SurgeryVideoQA and MIMICEchoQA were fully reviewed by clinical experts. Although YouTube explicitly prohibits PHI, we acknowledge residual risk and have added a clear warning in the paper.
>
> > Incremental insight — while scale is impressive, the core finding (“educational videos help”) feels somewhat intuitive and under-analyzed.
>
> We agree that the high-level intuition—that educational videos may contain useful biomedical information—is straightforward. However, our paper is fundamentally a dataset and benchmark contribution, with large scale data curation pipeline, diverse training dataset, and expert curated biomedical video evaluation benchmarks. Prior biomedical video resources are either narrow (e.g., echo-only, nursing-only, tool-tracking, consumer health) or too small to meaningfully test multimodal instruction tuning. In contrast, OpenBiomedVid spans 1,031 hours across anatomy, pathology, diagnostics, and surgery, making it by far the most diverse biomedical video-instruction dataset to date. Moreover, our two expert-curated benchmarks (MIMICEchoQA and SurgeryVideoQA) allow us to rigorously evaluate whether noisy, heterogeneous YouTube instructional content can yield models that generalize to clean, clinically curated distributions. Demonstrating this scale, diversity, and cross-domain transfer required building a new dataset, a new benchmarking suite, and running large multimodal models at multiple sizes.
>
> Response to Questions
>
> > Will the dataset and benchmarks be fully released? If so, under what license and with what access restrictions?
>
> Yes—both the dataset and benchmarks will be fully released. For the YouTube-sourced training set, we can only release metadata in accordance with YouTube’s Terms of Service; this includes the full instruction-tuning dataset (captions, Q/A pairs, and metadata), video URLs, and the exact clip timestamps, but not any video content or frames. Both expert-curated evaluation benchmarks (MIMICEchoQA and SurgeryVideoQA) will be released in full under a research-only, non-commercial license (CC-BY-NC), along with detailed documentation. We will also release all fine-tuned models under the same research-only license (CC-BY-NC), ensuring that the dataset, benchmarks, and models are openly available for academic use while respecting content ownership and privacy considerations.
>
> > How is personal or sensitive information (e.g., PHI, identifiable subjects) detected and filtered in the YouTube videos?
>
> We acknowledge the possibility that publicly available medical videos may contain identifiable individuals. To mitigate this, our pipeline includes multiple layers of automated and human filtering: (1) suspicious clips were manually reviewed during curation; and (2) all items in the evaluation benchmarks (MIMICEchoQA and SurgeryVideoQA) were fully reviewed by clinical experts, who removed or flagged any content containing facial imagery, names, on-screen identifiers, or other sensitive elements. Importantly, our released dataset contains only URLs and derived text annotations—we do not redistribute any video frames or audio. YouTube explicitly prohibits PHI, but we will nevertheless include a cautionary statement in the paper and documentation.
>
> > Does the fine-tuned model show transferability to unseen clinical or institutional datasets?
>
> Yes—the fine-tuned models demonstrate transferability to unseen clinical data. The MIMICEchoQA benchmark is sourced entirely from the MIMIC-IV-ECHO institutional dataset, not from YouTube, and is fully expert-curated, making it substantially cleaner and distributionally distinct from our training corpus. Despite this shift, both models show significant improvements: the 2B model improves by +99.1%, and the 7B model improves by +29.3% relative to their respective baselines.

---

> > ### Author Response · Authors · 2025-11-20
> > **Response Part 3**
> >
> > > What is the ratio of educational diagrams/narration-only videos to true clinical imaging?
> >
> > Approximately 39% of the dataset (≈ 9.5K videos, 400 hours) consists of clinical imaging modalities such as echocardiography, MRI, CT, X-ray, and microscopy. To quantify this more rigorously, we applied our SigLIP-based biomedical-frame classifier at 0.5 s resolution with a high-confidence threshold (>95% probability). Across the imaging subset, ~95% of sampled frames were predicted to be biomedical, providing a conservative estimate of imaging coverage. This automated analysis offers a reliable approximation of the proportion of true clinical imaging versus diagrammatic or narration-only content. We also note that a large portion of the remaining data—particularly surgical content—is not merely didactic animation or narration, but consists of real procedural footage captured in medical centers.
> >
> >
> > > Are there quantitative measures of data quality, such as annotation agreement or verification accuracy?
> >
> > Yes—our dataset includes several quantitative quality assessments. For the training corpus, we measured agreement on the biomedical–non-biomedical frame classification task by having two human annotators independently label SigLIP-selected frames. Human concordance with the SigLIP classifier was ≈95%, and inter-annotator agreement between the two humans was ≈93%, indicating that the automated filtering stage aligns closely with human judgment. For the caption-cleaning and QA-verification stages, as well as for constructing the evaluation benchmarks, we employed medical doctors: an expert cardiologist curated and validated the MIMICEchoQA benchmark, and a medical doctor spot-checked random samples of the training set and fully curated SurgeryVideoQA. Due to limited availability of domain experts, these were single-expert passes rather than multi-expert annotation studies, but all benchmark items were manually reviewed by clinicians to ensure high quality.
> >
> > To further quantify residual noise in our training dataset, we manually analyzed 100 randomly sampled caption–QA pairs. We found that 15% contained low-information or generic captions, 10% had mild grounding issues, and 5% exhibited LLM-like phrasing artifacts, while 70% were mostly clean and clinically grounded. This reflects the intended heterogeneity of the training corpus, which is drawn from naturally noisy YouTube educational videos. Importantly, both evaluation benchmarks are entirely expert-reviewed and do not contain these artifacts. We now include error-analysis section in the appendix.

---

> > > ### Author Response · Authors · 2025-11-28
> > >
> > > Dear reviewer,
> > >
> > > As the discussion period is nearing its end, we wanted to check whether our responses have sufficiently addressed your concerns, or if there are any remaining points that would benefit from further clarification. Thank you again for your time, thoughtful feedback, and engagement with our work.

---

### Author Response · Authors · 2025-11-20
**Overall Summary**

We thank all reviewers for their thoughtful and constructive feedback, and we appreciate the AC for coordinating the review process. Below, we synthesize and address the major concerns raised across multiple reviewers. We have also updated the manuscript with additional sections and result tables.

> 1. Evaluation Bias and Reliability of GPT-4o as a Judge

We appreciate the concern regarding potential stylistic or evaluator bias introduced by GPT-4o. To assess this, we conducted a comprehensive multi-judge analysis using both GPT-4o and Gemini-2.0-Flash across all 14 evaluated models. As shown in Supplementary Table S1, the two judges produce highly consistent accuracy estimates (overlapping 95% CIs), with 93.0% raw agreement and Cohen’s kappa = 0.748. Their model rankings are nearly identical (Spearman rho = 0.972, Pearson r = 0.993).

To validate judge reliability further, we ran a human evaluation on 400 items (Supplementary Table S4). Both GPT-4o and Gemini show strong alignment with human annotators (GPT-4o: 90–95%, kappa = 0.72–0.84; Gemini: 89–95%, kappa = 0.66–0.83), and agree with each other at 88–95% (kappa = 0.65–0.85).

Importantly, GPT-4o does not generate new content—the video captions originate from YouTube transcripts; GPT-4o only cleans and formats them into Q/A pairs. Every stage includes human verification, and the two benchmarks (SurgeryVideoQA and MIMICEchoQA) were entirely reviewed by clinical experts. Using LLMs for caption/Q&A generation is standard in prior work (e.g., LLaVA-Med, NeurIPS 2023; Video-ChatGPT, ACL 2024).

> 2. Lack of Deep Failure-Mode Analysis

In response to reviewer feedback, we performed quantitative and qualitative error analysis for both benchmarks. For MIMICEchoQA, errors concentrate in specific anatomical regions and ultrasound views (e.g., left atrium: 68.4%, pulmonary artery: 66.7%, PSAX/PLAX great-vessels: 70–71%). From 100 manually reviewed errors, ~60% involve severity miscalibration, 25–30% involve threshold ambiguities, and ~10% reflect systematic EF overestimation—patterns consistent with noisy clinical labels and borderline diagnostic categories.

For SurgeryVideoQA, manual analysis of 100 incorrect predictions reveals interpretable categories: anatomy/localization errors (22%), instrument/technique confusion (25–33%), procedural-step/temporal errors (16%), overly generic answers (10%), and quantitative misestimation (5–10%). These are mostly near-miss errors, reflecting the fine-grained surgical reasoning demanded by the benchmark.

> 3. Data-Quality Measures and Training-Set Noise Characterization

We provide multiple quantitative quality assessments. For frame filtering, two annotators independently labeled 1,000 SigLIP-selected frames, yielding 95% human–model agreement and 93% human–human agreement. Benchmark construction involved clinical experts: a board-certified cardiologist curated MIMICEchoQA and a medical doctor curated SurgeryVideoQA.

To further characterize training-set heterogeneity, we reviewed 100 randomly sampled caption–QA pairs. Approximately 15% contained low-information captions, 10% showed mild grounding issues, and 5% had LLM-style artifacts, while 70% were mostly clean and clinically grounded. This distribution reflects the intended heterogeneity of YouTube educational content that our paper investigates. Importantly, both evaluation benchmarks are fully expert-cleaned and do not contain these artifacts.

> 4. Comparison With Biomedical VLMs and Additional Video Baselines

We appreciate the suggestion to evaluate against medical VLMs. However, cited models such as LLaVA-Med, Med-Flamingo, and MedCLIP are single-image models and cannot process long video sequences (hundreds–thousands of frames) without major architectural modification.

We expanded Table 1 to include Video-ChatGPT and Video-LLaVA, and our original results already included competitive models capable of video understanding (e.g., InternVideo2.5-Chat-8B, Phi-3.5-vision-instruct, Phi-4-multimodal-instruct). These off-the-shelf video models achieve modest performance (e.g., 7.9–16.4% on SurgeryVideoQA), confirming that our benchmarks are genuinely challenging rather than tailored to Qwen architectures.

> 5. Training under constrained setting (e.g., fixed frame sampling, short 8–16 frame clips)

As shown in Table S3, we created a “biomed-frame” variant of our dataset by selecting the highest-confidence biomedical frames (via our SigLIP classifier) and pairing them with the same instruction-tuning Q/A. This provides a direct comparison between video-level and frame-level supervision on the same backbones. We find that frame-only tuning captures a substantial portion of the signal, however, full-video supervision remains superior. For instance, on Qwen2-VL-7B, MIMICEchoQA achieves 49.0% with video-level tuning vs. 44.0% with frame-level, and SurgeryVideoQA achieves 25.1% vs. 23.0%.

---

### Meta-Review · Area_Chair_ZCXj · 2025-12-29

**Summary:**

This paper introduces OpenBiomedVid, a 1,031-hour biomedical video instruction tuning dataset curated from YouTube educational videos, along with two expert-curated benchmarks (MIMICEchoQA and SurgeryVideoQA). The authors fine-tune Qwen2-VL models and demonstrate improvements on biomedical video and image benchmarks. The reviewers raised several significant concerns: (1) the pervasive use of GPT-4o throughout the pipeline—for caption cleaning, Q/A generation, metadata annotation, and evaluation—creates potential circular bias that human spot-checks on approximately 1% of training data cannot adequately address; (2) while the authors claim a 400-sample human adjudication study was conducted during rebuttal, this is inconsistent with Supplementary Table S4, which reports only 100 evaluated samples, and critical details about the human evaluation protocol remain unspecified; (3) the marginal advantage of video-based instruction tuning over frame-based approaches (only 5% absolute improvement in ablations) does not convincingly justify the paper's central claims about video-level supervision; (4) performance degradation on external benchmarks like MedQA (52.6% to 47.4% for the 7B model) contradicts the narrative that models are "learning medicine"; and (5) the self-constructed benchmarks share the same GPT-4o generation pipeline and YouTube domain as training data, raising concerns about whether improvements reflect genuine capability gains or distribution matching.

**Reviewer Concerns:**

Addressed concerns: The authors conducted a multi-judge analysis using both GPT-4o and Gemini-2.0-Flash, showing consistent rankings (Spearman ρ = 0.972) and reasonable agreement (κ = 0.748). This partially addresses Reviewer fKBP's concern about single-judge bias. The authors also provided detailed error analysis for both benchmarks, addressing Reviewer bp18's concern about lack of failure mode analysis. Additionally, the authors added comparisons with Video-ChatGPT and Video-LLaVA, partially addressing Reviewer fKBP's concern about underdeveloped baselines.

Outstanding concerns:

(1) regarding insufficient human verification of training data raised by Reviewers fKBP and bp18, the authors acknowledge that only approximately 1,000 samples (roughly 1.3% of 79,367 Q/A pairs) were spot-checked, and they "intentionally did not exhaustively clean the training set." While the authors claim a 400-sample human adjudication study was conducted during rebuttal, this is inconsistent with Supplementary Table S4, which reports only 100 evaluated samples. The discrepancy raises concerns about the rigor of the experimental methodology. Furthermore, critical details about the human evaluation—such as the number of annotators, their qualifications, and the evaluation protocol—remain unspecified.

(2) Marginal video vs. image advantage (Reviewer QWpu): The authors' own Table S3 shows frame-only tuning achieves 44.0% vs 49.0% (video) on MIMICEchoQA—only 5% improvement. This does not strongly support the claim that video-level supervision provides substantial advantages.

(3) MedQA performance degradation (Reviewer bp18): The authors acknowledge the drop but attribute it to "modality shift." However, this directly contradicts the paper's central claim about learning medicine. The improvement on PubMedQA does not fully compensate, as MedQA is more established for evaluating medical knowledge.

(4) Self-referential benchmark evaluation (Reviewers bp18, fKBP): Both benchmarks were generated using GPT-4o, and SurgeryVideoQA shares the YouTube domain with training data. While expert curation occurred, the strong improvements on self-constructed benchmarks (+99.1%, +52.1%) versus degradation on external MedQA suggests potential distribution matching rather than generalizable learning.

(5) Low absolute performance (Reviewer bp18): The 7B model achieves only 25.1% on SurgeryVideoQA—even GPT-4o achieves only 35.8%. The authors acknowledge this but frame it as justification for the benchmark's difficulty rather than addressing whether the Q/A quality is sufficient.

**Reviewer Scores:**

Reviewer QzV3, currently at 6, may maintain their score (guessed). The authors addressed most of their questions about data release, privacy, and quality measures, though the reviewer's core concerns about evaluation bias and ethical clarity were only partially addressed.

Reviewer bp18, currently at 4, would likely maintain their score or potentially lower it. The reviewer's primary concerns—GPT-4o bias creating evaluation-protocol-driven gains, low absolute performance, and MedQA degradation contradicting the "learning medicine" narrative—were not convincingly resolved. The authors acknowledged limitations but did not provide evidence that the improvements reflect genuine medical reasoning rather than stylistic alignment.

Reviewer fKBP (originally 2) indicated willingness to raise their score after the rebuttal. While some concerns were partially addressed, the score would likely increase only to 4, still below the acceptance threshold.

Reviewer QWpu, currently at 6, would likely maintain their score.

---

### Decision · Program_Chairs · 2026-01-26

Reject